psychology

belief in animal mind, animal purpose questionnaire, animal ethics, mind denial, animal use

**Author for correspondence:**
Helen J. Cassaday
e-mail: helen.cassaday@nottingham.ac.uk

# Man's best friends: attitudes towards the use of different kinds of animal depend on belief in different species' mental capacities and purpose of use

Matthew J. Higgs[1,2], Sasha Bipin[1] and Helen J. Cassaday[1]

[1]School of Psychology, University of Nottingham, University Park, Nottingham NG7 2RD, UK
[2]School of Medicine, Neuroscience and Mental Health Research Institute, Cardiff University, Cardiff, UK

(iD) HJC, 0000-0002-9227-373X

The animal purpose questionnaire (APQ) is a new instrument to measure human attitudes to animal use systematically across both species and purpose of use. This offers a more fine-grained approach to our understanding of how the belief in a specific animal's mental capacities relates to (dis-)agreement with their use for different human purposes. In the present study, 317 participants completed an online survey containing the APQ and the belief in animal mind (BAM) scale in a species-specific format, to test the prediction that levels of (dis-)agreement with animal use should mirror participants' judgements of animal sentience. The results obtained with the APQ confirmed that attitudes to animal use differed significantly across both purpose and species. Key findings included a relatively greater concern for dolphins and dogs over chimpanzees (suggesting that phylogenetic position is not the only determinant of attitudes to animal use). Across the purposes examined, respondents were largely negative about animal usage, with the exception that there was less disagreement if this was for medical research. Participants were also asked to provide demographic details such as gender and dietary preference. Regression analyses revealed high predictive power for species-specific BAM across five different kinds of animal use. General BAM scores, non-meat-eating and being female accounted for 31.5% of the total variability in APQ scores. The results indicate that BAM is a strong predictor of self-reported attitudes for using particular animals. However, the results showed some exceptions in the case of culturally typical 'produce' animals.

# 1. Introduction

Attitudes towards animal use are complex, often contradictory, and likely to be multiply determined. Nonetheless, the past few decades have seen considerable progress in mapping out the variety of factors that shape these attitudes. 'Animal use' refers to the human use of a variety of non-human animals (henceforth animals) for a variety of different purposes, and attitudes are 'the evaluation of an object, concept or behaviour along a dimension of favour or disfavour, good or bad, like or dislike' [1]. The most widely used measure for attitudes to animal use is Herzog *et al.*'s [2] animal attitude scale (AAS), recently shortened to a 10-item version [3], which generates a uni-dimensional score for attitudes to animal use. However, there is a wealth of evidence that an individual's attitude to the use of an animal would be dependent on factors such as what species the animal was [4–7] and what the purpose of the use was [6–9]. A number of the available scales have been used to examine how attitudes to animals depend on species [4] or purpose of use [9]. For example, Knight *et al.* [9] examined attitudes to different types of animal use (experimentation, teaching, personal decoration, entertainment, management/pest control and financial gain/food production) using a scale compiled from Armstrong & Hutchins [10] and Mathews & Herzog [11], as well as some additional questions. Knight *et al.* [12] advanced this line of work, examining attitudes towards the use of different kinds of animals for different kinds of purpose. However, rather than comparing individual participants' attitudes across species, each participant was asked about the use of only one type of animal across the selected purposes.

To our knowledge, the intersectionality between species and purpose has not been further addressed. In addition to employing conventionally formulated items such as 'It is morally wrong to hunt animals just for sport' [3], the present study also asked participants to indicate their attitudes to the use of a list of named animals (presented simultaneously to encourage comparative ratings), for a variety of specific purposes, in a structured and transparent format using the new animal purpose questionnaire (APQ). This approach allowed a systematic consideration of attitudes across selected species of animal and purposes of use, as well as assessment of convergent validity with the AAS.

Alongside species and purpose of use, several personal characteristics are known to account for variability in the attitudes people hold towards animal use. Age [13], sex [2,5], culture [14,15], vegetarianism [9], personality [11] and religion [16] have all been identified as important factors. The most powerful and consistent (negative) predictor of attitudes to animal use is belief in animal mind (BAM) [9,17–19].

A belief is considered 'the subjective probability that the object has a certain attribute' [1] and therefore, BAM describes an individual's belief regarding the existence of an animal's awareness, thoughts and feelings and their willingness to attribute mental capacities to an animal. Applying mental states to animals is considered 'commonplace, cross-cultural, species typical and almost irresistible' [20]. Hence BAM is often considered to emerge from knowledge based on direct daily experience or 'common sense' [18]. Hills [21] devised a four-item BAM scale, originally used to assess whether the public believed different classes of animals were capable of emotions, problem solving and awareness. This scale was further adapted to measure general BAM for 'most animals' [9], although the authors did acknowledge this was likely too broad a measure. It has also been reported that species is a key determinant of BAM [18,20,22], hence the current study employs a species-specific version of the BAM scale in place of the more general categories of animals examined in previous studies [9,21].

It is clear for both BAM and attitudes to animal use, that the species of animal in question is a key component to the moral consideration that animal receives, and various attempts have been made to understand how this relationship manifests. In experimental studies, participants rarely opt for a blanket or random attribution of mental ability to animals, but instead use a 'scala naturae' (phylogenetic) schema to organize the mental abilities of different species [20]. Herzog & Galvin [18] found the order of moral consideration to proceed as follows: invertebrates—fish—amphibians—reptiles—birds—mammals—dog/cat—primates. In philosophy, Singer [23] drew the line between the species that are entitled to moral consideration as lying somewhere between the shrimp and the oyster, but even within the classes of mammals, birds and reptiles, distinctions are drawn as to which animals are perceived as having a more complex or 'human-like' mental capacity [24].

If one believes that certain animals are mentally complex, then subjecting them to discomfort and death seems unacceptable [12] and hence a lack of mental complexity is often used to justify which species can be used and which uses are acceptable for that species. This link has been established in the literature [25,26], but the relationship is not clear-cut. For example, Knight *et al.* [12] found that scientists would willingly assign at least a 'moderate' level of sentience to typical research animals and yet remain heavily in favour of using animals for research. Thus, BAM alone does not necessarily predict what an individual

will advocate for animals. General BAM has been shown to predict attitudes to animal use, but a systematic comparison of species-specific BAM with species/purpose-specific uses will clarify how individuals navigate the complex moral issue of animal mental capacities and their utilization.

The present study used the APQ as part of an online survey to systematically compare the level of agreement for the utilization of 12 (groups of) animal species across five purposes. It was not our objective to sample species and purposes of use exhaustively, rather to test the validity of this approach to examining attitudes to animal use. Thus the selection of species/taxonomic groups was limited for the purposes of the present study in order to keep the overall survey completion time relatively short. Invertebrate species were not included because of the generally poor appreciation of their sentience among members of the general public. The particular animals selected were intended to be relatively familiar and unambiguous (from the perspective of participants' everyday understanding) and to span a variety of commonplace uses (including the categories of pet, pest and profit; [27]). The purposes of medical research (with some more immediate likely health benefit for humans, as well as potentially other animals) and basic science (with no immediate likely health benefit) were both included because this distinction is commonly drawn in connection with public understanding of animal research. Moreover, the legislation covering the research use of animals also draws on this distinction. For example, approvals of Project Licences by the UK Home Office require the completion of a harm-benefit analysis. Food production and pest control were included as benchmark purposes of use and additional uses were subsumed under the category of 'other'.

In addition, participants completed an adapted variant of Hills' [21] BAM scale for each of the 12 species/taxonomic groups to calculate species-specific BAM ratings (BAM-Species). To the authors' knowledge, there has yet to be any systematic assessment of participants' BAM scores for individual animals alongside their attitudes to different uses of that same animal. The present study, therefore, aimed to include a more systematic examination of the factors that predict attitudes to animal use, specifically species and purpose of use, as well as BAM-Species ratings for the same selected animals, taking into consideration the demographic characteristics of the participants as far as possible. Specifically, we had the objective to test for statistically robust differences in levels of concern by species and purpose, and to determine how any such identified differences relate to BAM and demographics such as gender and dietary preferences. Based on the evidence that females show higher levels of concern for animals [2,18,28] it was predicted that females would show generally higher levels of disagreement with animal use in the present study. Although the effects of diet were not the primary focus of the present study (and there was no specific recruitment strategy to sample non-meat-eaters), vegetarian and vegan participants were similarly expected to show higher levels of disagreement with animal use [9,25]. Other demographic questions requested that participants disclose their age, ethnicity, religion and level of education in order to determine the extent to which the sample was representative.

Species factors which might influence BAM include phylogenetic position. However, predictions based on phylogenetic placement taken in isolation are likely to be incorrect, since perceived bio-behavioural similarity to humans [24] as well as preconceptions based on familiar patterns of animal use [27] are also likely moderators of expressed attitudes. The present study, therefore, used a mix of hypothesis-driven and exploratory approaches. We have grounds to predict, and a sampling strategy suitable to test for, general effects of gender and diet but not the effects of other demographic variables (because of restricted variability or low N/demographic category). We do not set out to test directional predictions regarding the effects of phylogenetic position. As explained above, the list of species included was selective rather than exhaustive, and nor would we wish to claim that the particular selection of species included was definitive. Our objectives are rather to evaluate a 'blueprint' approach for asking questions about public attitudes to animal use as a function of species and purpose of that use, and to determine how such attitudes relate to BAM, gender and dietary preference.

# 2. Material and methods

The study was approved by the University of Nottingham School of Psychology Ethics Committee (Ref: 694R). The APQ scale is new and uses a within-subjects design to determine participants' ratings of different species, comparatively in the sense that they were required to rate a number of species, one after the other. Specifically, participants were asked to rate their agreement with the killing of each of the specific types of animals for each of the different purposes examined. A number of these animal uses would be unfamiliar if not offensive. Therefore, as a precautionary measure, participants were

advised that some of the animals mentioned in the study had never actually been used for the hypothetical purpose in the UK and that their ratings of agreement/disagreement were being taken in order for that data to be used for comparison purposes.

There were some generic examples of categories of purpose but there were no examples of how specific animals may be used in practice.

## 2.1. Participants and procedure

Three hundred and seventeen participants (223 females, five prefer not to say) aged between 18 and 80 years ($M = 38$ years, s.d. $= 15.98$ years) completed the online questionnaire. Further demographic details (other than age) are shown in table 1. Participants were recruited via a snowballing technique using friends, family and social networking sites. There was no financial incentive to participate.

Prior to beginning the questionnaire, the participants were presented with information and consent screens. They were informed that they would be presented with questionnaires asking questions about attitudes to animal use followed by the request for some background information, and advised of the likely study duration (15–20 min). They were reassured: that the information to be collected would be anonymous, to be used for research purposes only and stored in compliance with the Data Protection Act; that they could send any questions before, during or after the questionnaire to the contact email addresses provided; and that they could end their participation in the study at any time without providing a reason. Participants were required to indicate their consent before they could start. On completion of the survey, a debriefing statement explained the purpose of the study and the various measures that were used. Further reading was provided alongside contact details of the researchers, as well as support groups in case the participant experienced any distress.

## 2.2. Measures

The survey of questionnaires and demographic details was set up in Qualtrics [29] and comprised a series of sections.

### 2.2.1. Animal attitude scale

Participants first completed the short 10-item version of the AAS [3]. This scale is based on the original 20-item measure and includes items such as 'It is morally wrong to hunt animals just for sport'. Items are scored on a 5-point Likert scale: 1 = 'strongly disagree', 3 = 'undecided', 5 = 'strongly agree'. Items 2, 3, 4, 7 and 8 are reverse scored and overall higher scores correspond with greater animal welfare concern. Herzog *et al.* [3] reported an internal consistency of $\alpha = 0.90$. In the current sample, an internal consistency of $\alpha = 0.79$ was obtained.

### 2.2.2. Animal purpose questionnaire

The APQ consisted of five items per animal species to be used. Participants were asked to rate their level of agreement with the killing of each animal species for the following purposes: Medical Research, Basic Science Research, Food Production, Pest Control and 'Other'. The questionnaire instructions provided examples for each purpose of use: 'an animal model of dementia' as the Medical Research example; 'removal of that animal if it had damaged crops or invaded homes' as the Pest Control example; and 'hunting or fighting the animals as sport', 'wearing skin as fashion or as ornamentation' and 'for displaying the body as a trophy' as examples for 'Other'. For each animal species, participants were presented with a 5-item scale, ranging from 'strongly agree' to 'strongly disagree', for each of the five uses. Higher scores reflected stronger agreement with animal use and thus indicated relatively lower animal welfare concern. The presentation format used for the APQ is shown in appendix A.

The order of animals presented to each participant was randomized. The present study examined attitudes to 12 different types of animal species (Pig, Chicken, Dog, Dolphin, Chimpanzee, Rabbit, Rat, Snake, Frog, Pigeon, Fish, Parrot). The species were selected to provide a range of animals across the phylogeny as well as examples of animals that are traditionally considered pets, livestock and pests. Responses to the 5-item scale were coded 1 to 5 (as per the AAS) and the final raw scores were calculated as the average out of 5 (5 = highest agreement with animal use; hence lower APQ scores suggest pro-welfare attitudes) in relation to each of the purposes for each of the species. Average totals were calculated for each species, each purpose and overall. The APQ is primarily intended to provide

**Table 1.** Socio-demographic and lifestyle variables of respondents ($n = 272$–$317$). Items in bold were coded as the dummy variables for regression analyses (coded as 1, while the remaining items are coded as 0).

| | variable | % |
|---|---|---|
| gender | **female** | **70.3** |
| | male | 28.1 |
| | prefer not to say | 1.6 |
| ethnicity | **white** | **62.8** |
| | mixed/multiple ethnic backgrounds | 5.0 |
| | Indian | 24.6 |
| | any other Asian background | 2.8 |
| | any black background | 1.3 |
| | other | 0.6 |
| | prefer not to say | 2.8 |
| religion | **no religion** | **41.0** |
| | Christianity | 32.5 |
| | Judaism | 1.3 |
| | Islam | 1.9 |
| | Hinduism | 13.6 |
| | Buddhism | 0.6 |
| | Sikhism | 0.3 |
| | other | 3.5 |
| | prefer not to say | 5.4 |
| eating orientation | omnivore | 59.6 |
| | pescatarian | 4.4 |
| | flexitarian | 9.8 |
| | **vegetarian** | **15.1** |
| | **vegan** | **2.2** |
| | other | 3.2 |
| | prefer not to say | 5.7 |
| education | GCSE/equivalent | 6.9 |
| | A-level/equivalent | 14.8 |
| | **bachelor's degree** | **31.5** |
| | **master's degree** | **31.2** |
| | **doctorate** | **7.3** |
| | other | 4.7 |
| | prefer not to say | 3.5 |
| worked with animals | **yes** | **17.7** |
| | no | 82.3 |
| trained as a scientist | **yes** | **24.3** |
| | no | 75.7 |

species- and purpose-specific measures but the regression models also made use of the total scores. The internal consistency for this overall measure provided by the APQ was excellent ($\alpha = 0.982$), considered across the 60 items (12 species × 5 purposes). The APQ-Total is also suitable to check for

generally pro-welfare convergent validity with the AAS. The APQ-Total average score had a significant strong correlation with the AAS ($r = -0.757$, $p < 0.001$) confirming good convergent construct validity.

### 2.2.3. Belief in animal mind

BAM was measured using a customized version of Hills' [21] original four-item questionnaire to measure individual beliefs about animal awareness, thoughts and feelings. The original Hills' scale posed four questions in relation to four general categories of animal—mammals (non-human), fish, birds and insects; and subsequent studies have deployed the questionnaire with reference to 'most animals' [9]. In the present study, we used Hills' four questions for each of the 12 (groups of) species assessed in the APQ (*'To what extent do you agree that the following animal species are* unaware *of what is happening to them?'*; *'To what extent do you agree that the following animal species are capable of experiencing a range of feelings and emotions* (e.g. *pain, fear, contentment, maternal affection)?'*; *'To what extent do you agree that the following animal species are able to think to some extent, to solve problems and make decisions about what to do?'*; *'To what extent do you agree that the following animal species are more like computer programs* i.e. *mechanically responding to instinctive urges without awareness of what they are doing?'*). Participants were asked to rate their agreement with each statement on a 7-point Likert scale (ranging from 'strongly agree' to 'strongly disagree') for each of the 12 species before proceeding to the next item. In this format, participants could rate the species of animal comparatively for the same item. The presentation format used for the BAM-Species is shown in appendix B.

To control for response bias, two of the four items were negatively scored. Both the order of the four items and the order of the animal species within each item were randomized. Raw scores were in the range 1–7, with 7 equalling the highest possible BAM score a species could achieve. BAM was intended to be used as both a unitary measure of BAM and as a species-specific measure, hence internal consistency was measured for both. The internal consistency for the unitary measure was excellent ($\alpha = 0.958$), when calculated across the 48 items (12 species × 4 items). Internal consistency was more variable for the species-specific scales, ranging from satisfactory to good: Fish ($\alpha = 0.779$); Frog ($\alpha = 0.752$); Rat ($\alpha = 0.751$); Chicken ($\alpha = 0.724$); Pig ($\alpha = 0.716$); Pigeon ($\alpha = 0.708$); Snake ($\alpha = 0.697$); Rabbit ($\alpha = 0.676$); Dolphin ($\alpha = 0.658$); Parrot ($\alpha = 0.654$); Dog ($\alpha = 0.638$); Chimpanzee ($\alpha = 0.624$).

### 2.2.4. Demographics

The final section collected basic demographic information: gender, age, ethnicity, religion and the highest level of education. In addition, participants were requested to provide more specific information concerning their attitude to animals: eating orientation ('omnivore', 'pescatarian', 'flexitarian', 'vegetarian', 'vegan'); whether the participant had ever worked with animals; and whether the participant had trained as a scientist (the final two questions were answered dichotomously). These questions were asked at the end of the questionnaire to avoid priming participants' responses. Table 1 contains the distribution of participants within each demographic variable. The variables were coded as dummy variables for regression analyses in which the following responses were coded as 1: female, white, non-religious, having a degree, non-meat-eating (vegetarian and vegan), having worked with animals and having worked/trained as a scientist. All other responses were coded as 0.

## 2.3. Design and analyses

Two ANOVAs were conducted for APQ responses and for BAM-Species. Repeated measures designs were used for both. For the APQ there were two independent variables, Purpose (Medical Research, Basic Science Research, Food Production, Pest Control, Other) and Species (Pig, Chicken, Dog, Dolphin, Chimpanzee, Rabbit, Rat, Snake, Frog, Pigeon, Fish, Parrot). There was one dependent variable, the rating (1–5) of (dis-)agreement with animal use. Paired-samples *t*-tests with a Bonferroni correction were computed on all comparisons within the Purpose and Species factors. For the BAM-Species there was one independent variable, Species (Pig, Chicken, Dog, Dolphin, Chimpanzee, Rabbit, Rat, Snake, Frog, Pigeon, Fish, Parrot). There was one dependent variable, the rating (1–7) of agreement with animal sentience as measured by BAM. Paired-samples *t*-tests with a Bonferroni correction were computed on all comparisons within the Species factor. Bonferonni corrected Pearson's correlations were conducted for the relationships between demographic variables, APQ-Total and BAM-Total. To assess the predictive power of BAM, simple linear regressions were conducted to predict purpose-specific APQ attitudes using species-specific BAM, and a hierarchical

regression was conducted to predict total APQ attitudes from demographic variables and total BAM. For the regression analyses, it was also appropriate to collapse across species (to test how demographic factors might account for attitudes in relation to purpose) or purpose (to test how demographic factors might account for attitudes in relation to species). To test for differences by species for the demographic variables shown to be significant in the hierarchical regression, independent samples *t*-tests (with Bonferonni correction) were conducted to examine species-specific APQ and BAM-Species scores (for males versus females and meat-eaters versus non-meat-eaters).

The survey scales used bipolar adjectives, hence participants in effect rated semantic differential on an interval scale measuring levels of (dis-)agreement and parametric tests are in principle applicable [30–32] but see Liddell & Kruschke [33].

# 3. Results

The sample size and normality of the distribution provided further justification for the parametric approaches adopted. Mauchly's test of sphericity indicated that the assumption of sphericity was violated for Species, Purpose and Species × Purpose, hence Greenhouse-Geisser corrected values are reported for the below factorial analyses [34].

## 3.1. Factorial analysis of variance of the APQ responses

The two-way (5 × 12) repeated measures ANOVA of APQ responses revealed a significant main effect of Purpose $F_{2.24,706.12} = 113.77$, $MSE = 7.37$, $p < 0.001$, $\eta_p^2 = 0.265$, a significant main effect of Species $F_{6.27,1981.06} = 129.30$, $MSE = 2.10$, $p < 0.001$, $\eta_p^2 = 0.290$ and a significant Purpose × Species interaction $F_{19.44,6143.91} = 64.55$, $MSE = 0.68$, $p < 0.001$, $\eta_p^2 = 0.170$. The interaction was anticipated since the specific selection of animals examined included species that are traditionally used for specific purposes, such as livestock animals, and hence simple main effect analysis was considered unnecessary. Paired-samples *t*-tests within the Purpose factor (reported in table 2a) revealed significant differences between levels of agreement with animal use across all five purposes (all $ps < 0.019$) with one exception. There was no significant difference between support for animal use in Medical Research and Basic Science Research ($p = 0.142$). See figure 1a for mean APQ scores by purpose of use (re-scored so that zero reflects the neutral position).

Table 2b reports the Bonferroni-corrected pairwise comparisons by species in full. Due to the number of significant comparisons for the paired-samples *t*-tests on all comparisons within the Species factor, we will focus on the non-significant differences between species which reflect types of animals attracting similar levels of concern. The animal with the least agreement for its use was Dolphin ($M = 2.18$) but this was non-significantly different to Dog which had the second-lowest agreement ($M = 2.24$). Disagreement for Dog use was non-significantly different to Chimpanzee ($M = 2.29$) but Chimpanzee and Dolphin were significantly different from each other ($p = 0.027$). Parrot ($M = 2.45$) was the only animal for which ratings were significantly different to all other species (all $ps \leq 0.001$). There were five animals for which levels of endorsements of their use were all non-significantly different from each other: Pig ($M = 2.86$), Chicken ($M = 2.94$), Frog ($M = 2.87$), Pigeon ($M = 2.84$) and Snake ($M = 2.82$). Within this set of diverse animals, Snake was the only animal to be non-significantly different from Rabbit ($M = 2.72$), for which acceptance of use was lower than that seen for other animals in this grouping. At the upper extreme of the ratings, Chicken was the only animal found to be non-significantly different from Rat ($M = 2.98$), for which acceptance of use was higher than that seen for other animals in this grouping. Finally, Rat was also non-significantly different from Fish ($M = 3.06$) and Fish was significantly different from all other species (all other $ps \leq 0.001$). See figure 1b for mean APQ scores by species (re-scored so that zero reflects the neutral position).

## 3.2. Factorial analysis of the BAM responses

A one-way repeated measures analysis of variance of BAM-Species revealed a significant main effect of Species $F_{3.69,1164.83} = 312.05$, $MSE = 1.45$, $p < 0.001$, $\eta_p^2 = 0.497$. Due to the number of significant comparisons for the paired-samples *t*-tests on all species comparison, we will again focus on the non-significant differences between species. The BAM-Species for the following four animal species were significantly different from all others: Snake ($M = 4.59$), Frog ($M = 4.42$), Fish ($M = 4.22$) and Chimpanzee ($M = 6.27$). Fish was the animal with the significantly lowest BAM-Species and Chimpanzee was the animal with the significantly highest BAM-Species scores. The BAM-Species ratings for the following pairs of animals were found to be non-significantly different from each other: Dolphin ($M = 6.11$)

**Table 2.** The Bonferroni corrected pairwise comparisons for the APQ split by (a) purpose and (b) species. The left side of the table presents the difference between the means for the APQ scores for each species and each purpose. The right side of the table presents the p-values for those comparisons. Significant differences between the means are presented in bold.

| (a) purpose | medical | research | food | pest | other |
|---|---|---|---|---|---|
| Medical | — | 0.142 | <0.001 | <0.001 | <0.001 |
| Research | 0.068 | — | <0.001 | <0.001 | <0.001 |
| Food | **0.598** | **0.530** | — | 0.019 | <0.001 |
| Pest | **0.488** | **0.420** | **−0.110** | — | <0.001 |
| Other | **0.819** | **0.751** | **0.220** | 0.331 | — |

| (b) species | Pig | Chicken | Dog | Dolphin | Chimpanzee | Rabbit | Rat | Snake | Frog | Pigeon | Fish | Parrot |
|---|---|---|---|---|---|---|---|---|---|---|---|---|
| Pig | — | 0.470 | 0.000 | 0.000 | 0.000 | 0.001 | 0.042 | 1.000 | 1.000 | 1.000 | <0.001 | <0.001 |
| Chicken | −0.076 | — | <0.001 | <0.001 | <0.001 | <0.001 | 1.000 | 0.062 | 1.000 | 0.215 | 0.001 | <0.001 |
| Dog | **0.629** | **0.705** | — | 1.000 | 1.000 | <0.001 | <0.001 | <0.001 | <0.001 | <0.001 | <0.001 | <0.001 |
| Dolphin | **0.683** | **0.759** | 0.054 | — | 0.027 | <0.001 | <0.001 | <0.001 | <0.001 | <0.001 | <0.001 | <0.001 |
| Chimpanzee | **0.570** | **0.646** | −0.059 | 0.113 | — | <0.001 | <0.001 | <0.001 | <0.001 | <0.001 | <0.001 | 0.001 |
| Rabbit | **0.143** | **0.219** | **−0.486** | **−0.540** | **−0.427** | — | <0.001 | 0.985 | 0.003 | 0.008 | <0.001 | <0.001 |
| Rat | **−0.117** | −0.041 | **−0.746** | **−0.800** | **−0.687** | **−0.260** | — | 0.001 | 0.042 | 0.026 | <0.001 | <0.001 |
| Snake | 0.045 | **0.121** | **−0.584** | **−0.638** | **−0.525** | −0.098 | **0.162** | — | 1.000 | 1.000 | <0.001 | <0.001 |
| Frog | −0.008 | 0.068 | **−0.637** | **−0.691** | **−0.579** | **−0.151** | **0.109** | −0.054 | — | 1.000 | <0.001 | <0.001 |
| Pigeon | 0.021 | 0.097 | **−0.608** | **−0.662** | **−0.549** | **−0.122** | **0.138** | −0.024 | 0.030 | — | <0.001 | <0.001 |
| Fish | **−0.199** | **−0.123** | **−0.828** | **−0.882** | **−0.769** | **−0.342** | −0.082 | **−0.244** | **−0.191** | **−0.220** | — | <0.001 |
| Parrot | **0.415** | **0.490** | **−0.215** | **−0.269** | **−0.156** | **0.271** | **0.531** | **0.369** | **0.423** | **0.393** | **0.613** | — |

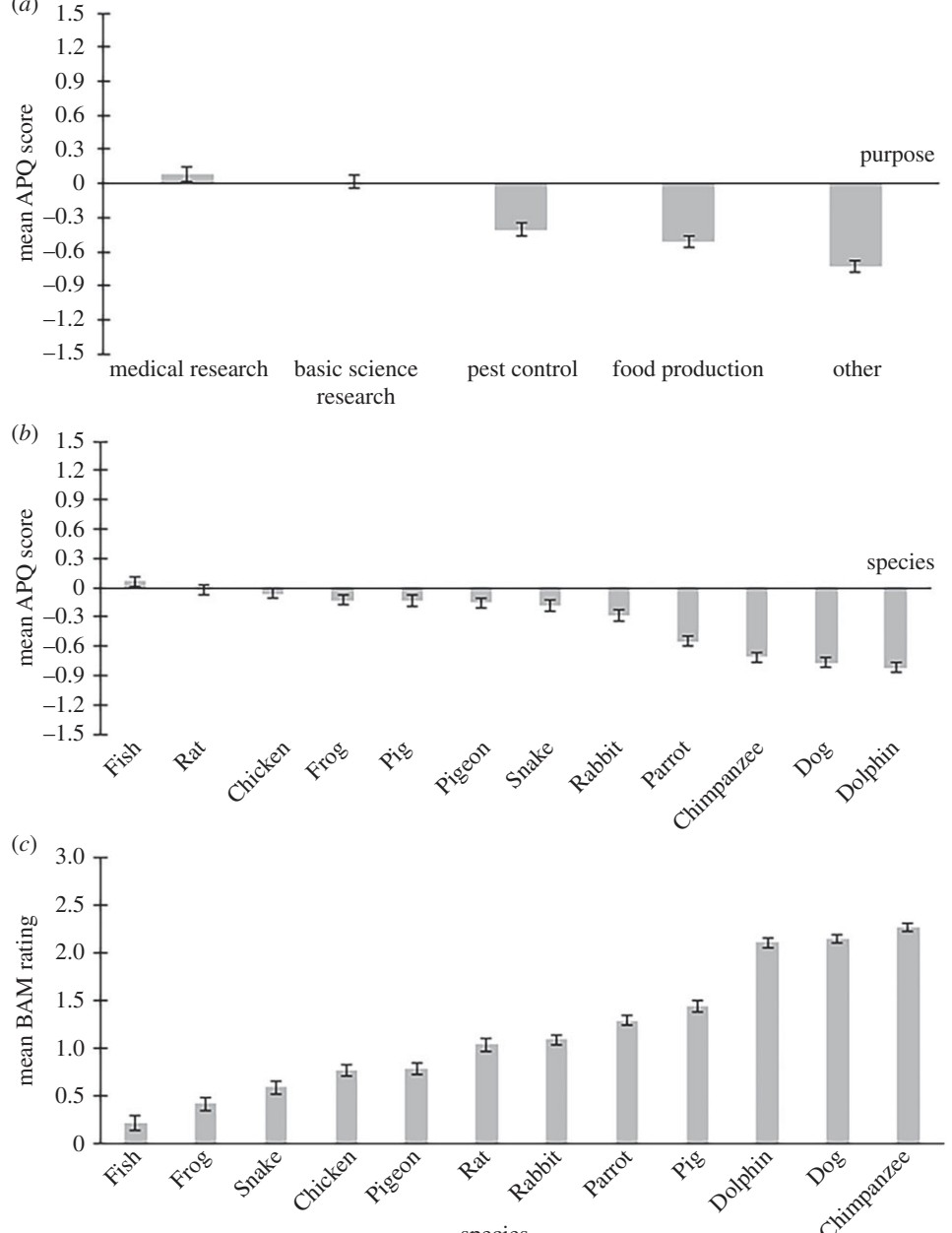

**Figure 1.** Mean scores and standard errors of responses from 317 participants from the different scales within the questionnaire. (*a*) displays the mean APQ scores when selecting for purpose (averaging across species), (*b*) displays the mean APQ scores when selecting for species (averaging across purpose) and (*c*) displays the mean BAM rating for each species. (*b*) and (*c*) are arranged in ascending order of animal welfare/concern. For the purposes of illustration, the APQ is shown re-scored from −2 to +2 (+2 = strongly agree with the animal use; hence lower scores indicate greater concern for animals). BAM ratings are re-scored from −3 to +3 (+3 = strongly agree with the animal's mental capacity). For both scales, zero represents a neutral position.

and Dog ($M = 6.12$); Pig ($M = 5.44$) and Parrot ($M = 5.29$); Rabbit ($M = 5.09$) and Rat ($M = 5.04$); and Chicken ($M = 4.77$) and Pigeon ($M = 4.79$). All other comparisons were significant ($p$s $< 0.006$). See figure 1*c* for mean BAM scores by species (re-scored so that zero reflects the neutral position).

## 3.3. Correlations between demographic variables, APQ scores and BAM scores

Gender was significantly correlated with APQ ($r = -0.315$, $p < 0.001$) as was eating orientation ($r = -0.390$, $p < 0.001$), with women and non-meat-eaters being less likely to support animal use. Eating orientation was also correlated with BAM ($r = 0.188$, $p = 0.001$), with non-meat-eaters being more likely to attribute animals with a higher mental capacity.

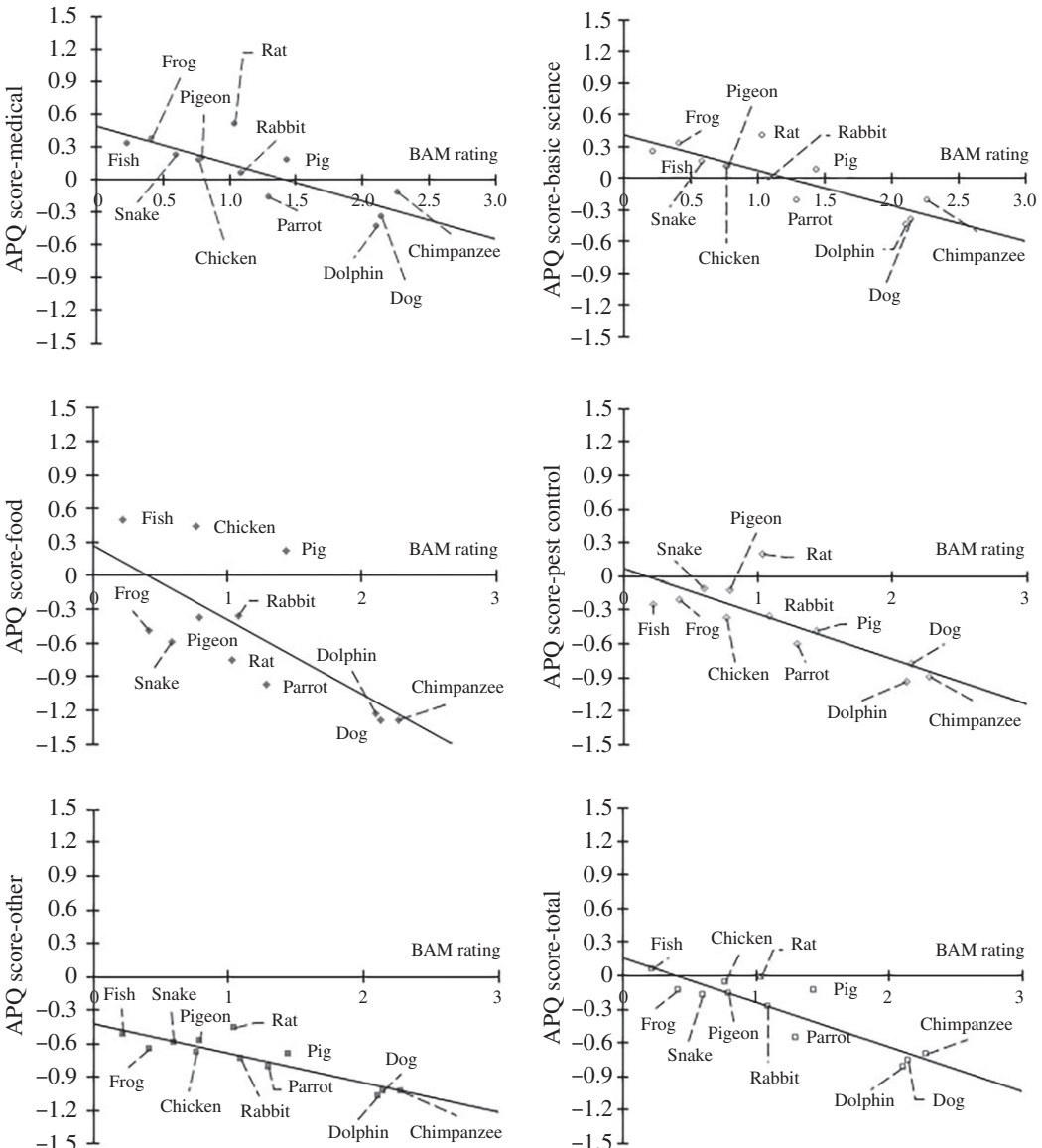

**Figure 2.** Mean APQ scores for each of the 12 species plotted against the specific BAM rating for that species. The mean ratings for Medical Research are displayed in the top left panel, Basic Science Research in the top right, Food Production in the middle left panel, Pest Control in the middle right panel, Other in the bottom left panel and the relationship between the APQ-Total score and BAM rating is present in the bottom right panel. Each plot contains a simple regression line highlighting the linear relationship present across all the different purposes. The mean APQ and BAM ratings were calculated from responses from 317 participants. For the purposes of illustration, the APQ is shown re-scored from −2 to +2 (+2 = strongly agree with the animal use; hence lower scores indicate greater concern for animals) and BAM ratings are re-scored from −3 to +3 (+3 = strongly agree with the animal's mental capacity). For both scales, zero represents a neutral position.

## 3.4. Simple linear regression analyses examining BAM as a predictor of APQ attitudes

Simple linear regressions were conducted to predict the purpose-specific APQ scores for each species using the average BAM-Species for each animal type. The sample size exceeded the necessary '104+m' rule of thumb, for the suggested minimum number of participants in relation to the number of predictors (m), for all regressions [35]. The final model was significant for the following purposes: Medical Research $F_{1,10} = 19.43$, $p = 0.001$, Basic Science Research $F_{1,10} = 23.06$, $p = 0.001$, Food Production $F_{1,10} = 10.49$, $p = 0.009$, Pest Control $F_{1,10} = 18.77$, $p = 0.001$, Other $F_{1,10} = 34.68$, $p < 0.001$ and APQ-Total score $F_{1,10} = 36.79$, $p < 0.001$. The $R^2$ indicated that the following percentage of the variability was accounted for by each model: Medical Research—65.9%, Basic Science Research—70.4%, Food Production—51.2%, Pest Control—65.6%, Other—77.7% and Total—78.6%. Figure 2 displays the linear relationships for the six regressions (re-scored so that zero reflects the neutral position).

**Table 3.** The $R^2$ and $\beta$ coefficients for the hierarchical regression of the demographic variables and BAM-Total on APQ-Total score.

|  | $\Delta R^2$ | $\beta$ |
|---|---|---|
| Step 1 | 0.110** |  |
| gender |  | −0.316** |
| age |  | 0.062 |
| ethnicity |  | −0.011 |
| Step 2 | 0.110** |  |
| religion |  | −0.018 |
| eating orientation |  | −0.303** |
| education |  | 0.079 |
| works with animals |  | −0.027 |
| scientist |  | 0.107 |
| Step 3 | 0.095** |  |
| BAM-Total |  | −0.322** |

**Significant at the 0.01 level.

## 3.5. Regression analysis of demographic variables and BAM as predictors of APQ attitudes

A hierarchical multiple linear regression was conducted with gender, age, ethnicity, religion, eating orientation, education, working with animals, trained as scientist and BAM-Total entered in steps and total APQ score as the dependent variable. Two hundred and seventy-two participants provided data for every demographic variable and this was sufficient to identify a medium effect size; there was no evidence of heteroscedasticity, multicollinearity or singularity (all $r$s < 0.9 and all tolerances >0.1). The final model was significant $F_{9,262} = 13.40$, $p < 0.001$ and accounted for 31.5% of the variability with an adjusted religion, eating orientation, education, working with animals and working as scientist were entered at step 2 and BAM-Total was entered at step 3. Overall, each step was significant with Step 1 accounting for 11% of the variability, Step two accounting for an additional 11% and Step 3 accounting for an additional 9.5%. The beta values indicated that, in Step 1, gender had a significant negative relationship to APQ scores, showing that women were more likely to disagree with animal use than men. In Step 2, only 'eating orientation' was significantly negatively related to APQ scores, meaning that vegetarians and vegans were more likely to disagree with animal use and Step 3 found BAM was significantly negatively related to APQ scores, hence the more an individual believes in animal mind, the less likely they are to agree to animal use. Table 3 contains $R^2$ and $\beta$ coefficients for each step of the regression.

## 3.6. Independent samples *t*-tests examining APQ and BAM-species for gender and eating orientation

Independent samples *t*-tests with Bonferonni correction were conducted for the sub-populations: male and female; meat-eating and non-meat-eating (table 4). APQ scores were significantly different for all species between males and females, with females less likely to support animal use (all $p$s < 0.001) and APQ scores were significantly different for all species between meat-eaters and non-meat-eaters, with non-meat-eaters less likely to support animal use (all $p$s < 0.001). BAM ratings did not differ significantly between males and females for any species but there was a significant difference between meat-eaters and non-meat-eaters for BAM-Fish ($p = 0.001$), with non-meat-eaters attributing higher mental capacities to this animal.

## 4. Discussion

The advantage of the APQ over other scales is that it allows direct comparisons between species within categories of purpose of use. The scale is transparent and implemented without any reverse scoring: the

**Table 4.** The means and standard deviations of (a) APQ and (b) BAM (raw scores) for each species for the sub-populations: male and female, and non-meat-eating and meat-eating. Each set of values is accompanied by the outcome of independent samples t-tests conducted between the sub-populations. Significant values after Bonferroni correction are highlighted in bold.

| species | female M | female s.d. | male M | male s.d. | t | p | meat-eating M | meat-eating s.d. | non-meat-eating M | non-meat-eating s.d. | t | p |
|---|---|---|---|---|---|---|---|---|---|---|---|---|
| **(a) APQ** | | | | | | | | | | | | |
| Pig | 2.69 | 0.96 | 3.31 | 0.90 | −5.25 | <0.001 | 2.08 | 1.07 | 3.07 | 0.84 | −6.38 | <0.001 |
| Chicken | 2.78 | 0.93 | 3.38 | 0.86 | −5.28 | <0.001 | 2.10 | 0.97 | 3.15 | 0.81 | −7.53 | <0.001 |
| Dog | 2.03 | 0.87 | 2.76 | 1.02 | −6.40 | <0.001 | 1.74 | 0.83 | 2.35 | 0.96 | −4.34 | <0.001 |
| Dolphin | 2.04 | 0.84 | 2.54 | 1.00 | −4.46 | <0.001 | 1.76 | 0.91 | 2.28 | 0.88 | −3.91 | <0.001 |
| Chimpanzee | 2.15 | 0.85 | 2.68 | 0.94 | −4.79 | <0.001 | 1.83 | 0.83 | 2.40 | 0.89 | −4.31 | <0.001 |
| Rabbit | 2.54 | 0.94 | 3.22 | 0.95 | −5.74 | <0.001 | 2.04 | 0.99 | 2.90 | 0.92 | −6.20 | <0.001 |
| Rat | 2.82 | 0.99 | 3.40 | 0.77 | −5.46 | <0.001 | 2.25 | 1.05 | 3.18 | 0.84 | −6.19 | <0.001 |
| Snake | 2.67 | 0.99 | 3.22 | 0.84 | −4.55 | <0.001 | 2.14 | 1.03 | 3.00 | 0.88 | −6.33 | <0.001 |
| Frog | 2.73 | 0.93 | 3.25 | 0.83 | −4.59 | <0.001 | 2.12 | 1.00 | 3.07 | 0.80 | −6.63 | <0.001 |
| Pigeon | 2.68 | 0.98 | 3.28 | 0.86 | −5.03 | <0.001 | 2.17 | 1.02 | 3.03 | 0.89 | −5.79 | <0.001 |
| Fish | 2.91 | 0.94 | 3.47 | 0.82 | −4.93 | <0.001 | 2.31 | 1.04 | 3.26 | 0.80 | −6.44 | <0.001 |
| Parrot | 2.32 | 0.89 | 2.80 | 0.92 | −4.26 | <0.001 | 1.87 | 0.85 | 2.59 | 0.88 | −5.51 | <0.001 |
| total | 2.53 | 0.80 | 3.11 | 0.76 | −5.84 | <0.001 | 2.03 | 0.86 | 2.86 | 0.73 | −6.61 | <0.001 |
| **(b) BAM** | | | | | | | | | | | | |
| Pig | 5.44 | 1.15 | 5.43 | 0.99 | 0.23 | 0.821 | 5.70 | 1.30 | 5.39 | 1.04 | 1.66 | 0.101 |
| Chicken | 4.86 | 1.21 | 4.55 | 1.09 | 2.35 | 0.019 | 5.24 | 1.41 | 4.64 | 1.10 | 2.92 | 0.005 |
| Dog | 6.25 | 0.81 | 5.91 | 1.00 | 2.89 | 0.004 | 6.37 | 0.90 | 6.13 | 0.84 | 1.92 | 0.056 |
| Dolphin | 6.14 | 0.92 | 6.01 | 0.94 | 1.14 | 0.256 | 6.30 | 0.99 | 6.09 | 0.89 | 1.59 | 0.114 |
| Chimpanzee | 6.33 | 0.81 | 6.14 | 0.92 | 1.85 | 0.065 | 6.46 | 0.85 | 6.27 | 0.80 | 1.59 | 0.130 |
| Rabbit | 5.18 | 1.05 | 4.86 | 1.02 | 2.75 | 0.006 | 5.44 | 1.17 | 5.01 | 1.00 | 2.52 | 0.014 |
| Rat | 5.13 | 1.22 | 4.82 | 1.18 | 2.27 | 0.024 | 5.31 | 1.39 | 4.98 | 1.17 | 1.65 | 0.104 |
| Snake | 4.67 | 1.23 | 4.40 | 1.05 | 2.15 | 0.032 | 4.98 | 1.42 | 4.48 | 1.09 | 2.45 | 0.017 |
| Frog | 4.51 | 1.31 | 4.20 | 1.10 | 2.21 | 0.028 | 4.86 | 1.53 | 4.30 | 1.15 | 2.57 | 0.012 |
| Pigeon | 4.85 | 1.16 | 4.64 | 1.02 | 1.70 | 0.090 | 5.17 | 1.31 | 4.68 | 1.05 | 2.58 | 0.012 |
| Fish | 4.29 | 1.37 | 4.07 | 1.22 | 1.66 | 0.098 | 4.80 | 1.46 | 4.07 | 1.25 | **3.45** | **0.001** |
| Parrot | 5.37 | 1.01 | 5.11 | 0.96 | 2.12 | 0.035 | 5.65 | 1.05 | 5.22 | 0.96 | 2.92 | 0.004 |
| total | 5.25 | 1.10 | 5.01 | 1.04 | 2.44 | 0.015 | 5.52 | 1.07 | 5.11 | 0.79 | 2.74 | 0.008 |

participants' comparative judgements are the main focus. The results obtained with the new APQ confirmed that attitudes to animal use differed significantly across purpose and species.

The disagreement with using animals and the BAM-Species ratings for different species seemed (at least in part) to follow judgements of phylogenetic position; the relationship that previous studies have found (e.g. [20]). The present study included controls for response bias (negative scoring and randomized order of items and species) but we cannot exclude the possibility that the inherently comparative format of the scales may have influenced participants to vary their responses by species [26]. Notably, however, and despite any demand characteristics to express generally pro-welfare attitudes, we find consistent patterns in the levels of disagreement with animal use. The category of Fish species was the only one with a mean average score reflecting some level of agreement with use across all of the examined purposes. Similarly, with respect to the purposes of use examined in the present study, respondents were largely negative about animal usage across the board, with medical research being the only exception (and with a somewhat neutral score, suggesting ambivalence rather than positivity). The overall attitude to use of animals in basic science research was neutral; but there was no significant difference between levels of agreement with animal use in basic science versus medical research. While we might assume that there should be relatively higher endorsement of the use of animals in medical research, systematic evidence on this point is lacking. Vegetarians might see medical research as a more essential use of animals than eating them, but appreciation of the importance of basic science research might depend on level of education. This was measured in the present study but not examined in the sense that the majority of the sample was highly educated (as shown in table 1). All other differences between the purposes of use survived Bonferroni correction.

Consistent with the generally pro-welfare attitudes expressed in the APQ, BAM ratings were above neutral for all of the species examined reflecting variable levels of belief (as opposed to disbelief) in animal mind. Moreover, with a few exceptions, the BAM-Species showed that higher levels of belief in the mental capacities of particular species were typically related to less support for using that animal across the range of purposes.

Overall, BAM, alongside gender and eating orientation, explained 31.5% of the variability in an individual's total APQ scores. Previous studies have found that not eating meat accounts for a small proportion of the variance in attitudes to animal use [9,25] but a further relationship between not eating meat and higher BAM was found in this study. Knight *et al.* [9] suggested that higher levels of BAM do not lead to not eating meat (nor vice versa). However, in our dataset, eating orientation had a significant correlation with BAM-Total and we found that out of all the species, Fish had a significantly higher average BAM-Species for non-meat-eaters (not including pescatarians) after a very conservative Bonferroni correction. Taken as a whole, the results shown in table 4 suggest that participants who have chosen not to eat meat adjust their baseline rankings of sentience such that all animals are seen as more mentally capable, while maintaining the same perception of phylogenetic position as meat-eaters. Religious and ethnic identities are also likely determinants of attitudes to animal use, in part because of cultural variation in which animals are considered appropriate foods. In the present study ethnicity and religion were not identified as significant predictors (table 3). However, as shown in table 1, while reasonably representative of the UK population the sample was 62.8% white and largely Christian (32.5%) or disclosed no religion (41%). Cross-cultural studies will be necessary to examine the effects of these demographics in more diverse samples.

One of the most interesting findings in the present study is the large gender difference in the APQ scores for the 12 types of species compared to the more modest gender differences in the BAM-Species scores. Gender differences are well established in the literature [2,18,28] for both BAM and attitudes to animal use. However, although females were found to have a stronger disagreement for using every species of animal compared to males in this study, this was not matched by a corresponding increase in BAM for any of the species. Very little progress has been made in finding the underlying explanation for gender differences [36] but these data suggest BAM is unlikely to play a large role (if any) in why men and women differ in their attitudes to animals. Similarly, with the exception of the ratings for Fish (discussed below), the robust differences seen between the meat-eaters and the non-meat-eaters for the APQ ratings were not reproduced in the BAM-Species ratings.

These demographic details aside, at the group level, regression analyses demonstrated a strong linear relationship across all purposes of use, so that the higher the BAM-Species for a particular animal type, the less likely it was to be supported for use. However, within the category of purpose Food Production, BAM-Species failed to have as much predictive power. Participants reported a lower acceptance for the use of species which scored lower on BAM-Species but are not usually eaten by UK populations. Conversely, the endorsement of eating Pig, Chicken and Fish was beyond what was predicted by their BAM-Species. Part of this discrepancy appears to be that species that were scoring a lower BAM-Species score than would be predicted by the phylogenetic pattern, such as Pig compared to other

large mammals; Chicken and Pigeon compared to Parrot; and Fish compared to Frog and Snake. Such mismatch may result from the 'mind denial' effect described by Bastian *et al*. [25].

'Mind denial' (in the sense of failing to appreciate animal sentience) is a form of cognitive dissonance in relation to meat consumption [25,26,37]. It is a common occurrence that many people endorse the benefits of using an animal but most, when in a situation to, are reluctant to harm things that have minds [25]. This creates the state of cognitive dissonance—an uncomfortable inconsistency between behaviours and attitudes [38], and escaping this dilemma is a strong motivator for individuals to deny that animal species they eat have complex minds [6]. By regarding animals as automata, we can effectively place them outside of moral consideration [39] and conceive of them as an out-group [26]. Consistent with a role for mind denial, in the present study, the BAM-Fish ratings were significantly different between meat-eaters and non-meat-eaters. Most members of the public are exposed to the use of animals in food consumption on a daily basis. For those who consume animal products, perceived limitations to the minds of the animals they consume help to overcome cognitive dissonance. Non-meat-eaters no longer experience any such conflict in relation to their appreciation of other animals' mental capacities. However, the pattern of responses seen to Pig is inconsistent with this interpretation; eating pigs was deemed comparatively acceptable by respondents despite relatively high BAM-Pig ratings. Moreover, the BAM-Pig ratings were not different between meat-eaters and non-meat-eaters. So, meat-eaters and non-meat-eaters alike believe in pig minds, but meat-eaters still eat them.

Relatedly, the interaction between species and purpose for the APQ is confounded by familiarity effects in relation to specific species for specific purposes (particularly for diet). Feelings of disgust in relation to the prospect of eating certain species could result from belief in animal sentience or other features of the animal (such as being slimy in the case of frogs or carriers of disease in the case of rodents). Other cultural factors are also likely to be important. Eating pigs is widely accepted in the UK. Different attitudes to eating pork prevail in countries where Islam is the dominant religion. Nonetheless, systematic data on differences in public attitude by species and purpose can help us to understand inconsistencies in human behaviour. For example, animal welfare activists actively campaign against the use of laboratory rodents, yet their routine extermination in the course of pest control attracts relatively little attention. The APQ provides a tool to measure how this practical distinction relates to reported attitudes.

Similarly, the comparatively high BAM-Dog ratings (similar to Dolphin and Chimpanzee), in this case coupled with general disagreement with use across the purposes examined, likely reflect the special status of dogs in a population for which dogs are such a popular pet (as opposed to a potential nuisance or stray animal, or even a food, as may be the case in other countries not reached by the sampling strategy adopted in the present study). The relatively greater concern for dolphins and dogs over chimpanzees for example is of interest because primates might be expected to receive the greatest moral consideration based on phylogenetic position [18,20,24]. In terms of underlying mechanisms, these deviations in the ratings (from what would be predicted based on phylogeny alone) suggest that BAM-Species is rather related to an animal's perceived behavioural as well as biological similarity to humans [24]. Hence dolphins and chimpanzees which are perceived as socially and cognitively complex species were more highly rated than the other mammals sampled. Similarly, parrots (commonly perceived as intelligent because of their capacity to mimic human speech) were rated above pigeons and chickens.

Despite these 'outliers', the results confirm the utility of the BAM as a predictor of attitudes to animal use. Because such outliers can be identified, the findings also confirm the usefulness of asking the BAM questions in relation to particular named animals rather than of 'most animals' in general. Comparison of BAM-Species and attitudes towards the use of that species helps us to delineate how public views of animal complexity relate to the willingness to use specific animals. Systematic comparisons of BAM-Species and animal use are, therefore, suggested as the standard for future research. The selection of species/taxonomic groups was restricted to 12 exemplars for the purposes of the present study in order to keep the overall survey completion time relatively short. The particular animals selected were intended to be relatively familiar and unambiguous (from the perspective of participants' everyday understanding) and to span a variety of commonplace uses (including the categories of pet, pest and profit; [27]). Invertebrates were not included in the present study because of the limited understanding of their sentience among members of the general public. However, we recommend that invertebrates be considered for future deployments of the scale. Future larger scale uses of the new APQ scale could also benefit from selecting species identified with further animal groupings in a nested structure, perhaps based on phylogeny, to benefit from cluster analysis [40] and/or multilevel modelling approaches [41].

A further limitation was the potential conflation of diverse species in some of the listed animal types. Participants' appreciation of differences in sentience between smaller fish species is likely limited and 'fish' are typically treated as an entity for dietary purposes (e.g. participants may describe themselves as fish-eating vegetarians or pescatarians). However, we would expect different levels of concern for shark species

of fish. The APQ is readily adaptable for use in future studies, to measure attitudes to the use of a wider range of more specifically identifiable species for different purposes. In the present study, we focused on a few broad categories of animal use and the focus was the consequence that the animals would (ultimately) be killed. We cannot assume that the same pattern of results would be seen if participants were instead directed to consider other areas of animal use which do not result in the animals' early death (e.g. for dairy products or wool production; for transport or carrying; as companion animals). This too remains to be tested. Similarly, the category of 'other' was somewhat simplistic and relatively higher levels of disagreement with the use of animals for ill-defined other purposes could relate to the ambiguity and/or 'catch all' nature of this category. Notwithstanding these limitations, we propose that the survey used in the present study provides a useful template for future work to address further distinctions of interest. For example, the comparison between the purposes of basic science and medical research would be of particular interest following activities intended to promote public understanding of animal research.

Moreover, the APQ may also provide a useful tool, to further investigate attitudes to animal use by species and purpose in particular populations such as scientists and animal welfarists [12]. The newly developed Speciesism Scale [4] will also provide valuable insights into the origins of population differences, including those likely to be evident cross-culturally. This theoretically driven scale has been empirically validated and shown to provide a reliable measure of prejudice against animals, which is both stable within individuals and systematically related to other differences between individuals, from social dominance to other forms of prejudice such as racism. However, this scale has been devised to measure a general construct rather than as a tool to distinguish differences in attitudes in relation to purpose and species.

As suggested by Knight *et al*. [12], future research in the area of BAM should also adopt designs which will allow causal conclusions about the role of BAM in deciding attitudes towards animal use. Surveys conducted pre- and post-public engagement activities which will likely give valuable information on this point. Alternatively, participants' beliefs in animal sentience could be experimentally manipulated through the use of vignettes. Although Hills' scale is useful, more accurate measures of animal mind may be acquired through providing ecological examples of animal behaviour [42] and/or by distinguishing the primary and secondary emotions which may be attributed to animals [43]. There also needs to be a stronger focus on qualitative analysis to complement quantitative findings. As Knight & Barnett [6] reflected, the relationship between factors and attitudes is not always based on rational consideration of the relevant factors, hence it is naive to expect this relationship to be rigid and predictable. However, the present study has demonstrated that the topic of attitudes to animal use and BAM benefits from a systematic approach across both species and purpose, and that BAM-Species can have influence over the attitudes people possess towards different kinds of animals, and how these attitudes vary across different categories of use.

Research ethics. The study was approved by the University of Nottingham School of Psychology Ethics Committee (Ref: 694R). Participants were required to indicate their informed consent before they could start the survey.

Data accessibility. The raw data is deposited in the University of Nottingham Research Data Repository (https://rdmc. nottingham.ac.uk/) and freely available post-publication (doi:10.17639/nott.7035).

Authors' contributions. All authors developed the study concepts and contributed to the study design. Testing and data collection were performed by M.J.H and S.B. M.J.H. performed the data analysis and interpretation under the supervision of H.J.C. M.J.H. drafted the manuscript and H.J.C. provided critical revisions. All authors approved the final version of the manuscript for submission.

Competing interests. M.J.H., S.B. and H.J.C. have no competing interests.

Funding. We received no funding for this study. This work reported in this manuscript was the Masters projects of M.J.H. and S.B.

Acknowledgements. We thank Sarah Jones for proofreading the manuscript.

# Appendix A

The Qualtrics presentation format for the animal purpose questionnaire (APQ). The question 'To what extent do you agree with the use of…' was repeated in the identical format for 12 types of animal. Thus participants were asked to repeat the ratings also for CHICKEN, DOG, DOLPHIN, CHIMPANZEE, RABBIT, RAT, SNAKE, FROG, PIGEON, FISH and PARROT (just the animal to be considered was changed in the question line). In practice, the precise order in which participants were asked to consider different animals was randomized.

You will now be asked to rate whether you agree or disagree with the killing of different types of animal for the following purposes:

*medical research* (of any kind e.g. for an animal model of dementia)

*basic science research* (of any kind e.g. to better understand the brain)

*food production* (e.g. any form of commercial or domestic consumption of animal meat)

*pest control* (e.g. removal of that animal if it had damaged crops or invaded homes)

*other* (e.g. for hunting or fighting the animals as a sport; for wearing skin as fashion or as ornamentation; for displaying the body as a trophy)

Select the appropriate option on the rating scales to indicate your level of agreement or disagreement with the use or treatment of animals, within each of these broad categories, which directly or indirectly results in the killing of the animal.

Even if the use or treatment of the particular animal seems unlikely, imagine it was a typical use of the animal and make the judgement accordingly.

○ Check this box to continue

To what extent do you agree with the use of a **PIG** for the following purposes:

| | strongly agree | agree | neutral | disagree | strongly disagree |
|---|---|---|---|---|---|
| medical research | ○ | ○ | ○ | ○ | ○ |
| basic science research | ○ | ○ | ○ | ○ | ○ |
| food production | ○ | ○ | ○ | ○ | ○ |
| pest control | ○ | ○ | ○ | ○ | ○ |
| other | ○ | ○ | ○ | ○ | ○ |

# Appendix B

The Qualtrics presentation format for the Belief in Animal Minds (BAM) questions. The same format was used for the other 3 BAM questions: *'To what extent do you agree that the following animal species are capable of experiencing a range of feelings and emotions (e.g. pain, fear, contentment, maternal affection)?'*; *'To what extent do you agree that the following animal species are able to think to some extent, to solve problems and make decisions about what to do?'*; *'To what extent do you agree that the following animal species are more like computer programs i.e. mechanically responding to instinctive urges without awareness of what they are doing?'* In practice, the order of the four items was randomized and the order of the animal species within each item was also randomized.

To what extent do you agree that the following animal species are *unaware* of what is happening to them?

| | strongly agree | agree | somewhat agree | neither agree nor disagree | somewhat disagree | disagree | strongly disagree |
|---|---|---|---|---|---|---|---|
| Pig | ○ | ○ | ○ | ○ | ○ | ○ | ○ |
| Chicken | ○ | ○ | ○ | ○ | ○ | ○ | ○ |
| Dog | ○ | ○ | ○ | ○ | ○ | ○ | ○ |
| Dolphin | ○ | ○ | ○ | ○ | ○ | ○ | ○ |
| Chimpanzee | ○ | ○ | ○ | ○ | ○ | ○ | ○ |
| Rabbit | ○ | ○ | ○ | ○ | ○ | ○ | ○ |
| Rat | ○ | ○ | ○ | ○ | ○ | ○ | ○ |
| Snake | ○ | ○ | ○ | ○ | ○ | ○ | ○ |
| Frog | ○ | ○ | ○ | ○ | ○ | ○ | ○ |
| Pigeon | ○ | ○ | ○ | ○ | ○ | ○ | ○ |
| Fish | ○ | ○ | ○ | ○ | ○ | ○ | ○ |
| Parrot | ○ | ○ | ○ | ○ | ○ | ○ | ○ |

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
