## [Reviewer comments · Royal Society Open Science]

Review History

RSOS-181697.R0 (Original submission)

Review form: Reviewer 1

Is the manuscript scientifically sound in its present form?

Yes

Are the interpretations and conclusions justified by the results?

Yes

Is the language acceptable?

Yes

Is it clear how to access all supporting data?

No

Do you have any ethical concerns with this paper?

No

Have you any concerns about statistical analyses in this paper?

No

Recommendation?

Reject

Comments to the Author(s)

This is an interesting paper that is trying to achieve a reasonable goal – to identify more specifically the intersectionality of species and purpose in terms of attitudes towards animals. It is indeed true that people's attitudes towards animals, and how they should be used, varied widely. That said, it is also this variation and idiosyncratic approach to how people think about animals, which makes the goal of establishing a measure to capture this variance a somewhat difficult goal. For one thing, it suggests that such a measure should be endlessly complex (incorporating an exhaustive list of animals or purposes) if it is to map the territory. It was perhaps this question about the current paper – what the purpose of this measure is supposed to be, and what variance it is exactly measuring – which remained unclear for me and therefore reduced my enthusiasm. A few more specific points below:

1. The selection of uses seem somewhat arbitrary to me. It was unclear why medical research and basic science research should be separated out, yet use for clothes and fighting and hunting collapsed into the category 'other'.
2. The selection of animals was explained, but again my question here is what is the goal of this measure? It seemed that, in the end, it was to show that variation exists, rather than to try and pinpoint any specific attitudes about specific species. But then, I wonder what one would do with this measure, except to show that variation exists? For instance, I was very unsure what to make of the finding that there was low level agreement on purpose of use for Dolphins and Dogs? Why is this and what does it tell us?
3. The authors use the BAM to predict attitudes towards uses of different species of animals, but also acknowledge that much of the variance here may be to do with social norms – such as which animals we eat. For instance, attitudes towards eating rats is unlikely primarily determined by their perceived mental capacities. This immediately throws up the problem that the APQ is capturing all sorts of variance and this varies as a function of species and purpose, making it unclear what the instrument is measuring.
4. Some of the analyses focused on collapsing the APQ, which lead me to wonder how these analyses related to the species-specific ones.

In the end, I do think that the general aim of the paper is worthy, but I just could not clearly see what to take away from the data, or how this measure would be useful. To summarise my take, it seemed like the measure revealed that there is disagreement that animals should be used for any purpose, that there is more disagreement about some animals than others, and the belief in animal mind appears to explain whether it is considered okay in general to use an animal for a purpose. I did not find the collapsed analysis insightful, or showing anything unique to the APQ, given it was designed to investigate differences between animal and purpose, rather than collapse across them. Perhaps I have missed something, and if so then perhaps the authors could consider how they might bring out the unique contribution and purpose of this measure a little more in their explanation and treatment of it.

Review form: Reviewer 2**Is the manuscript scientifically sound in its present form?**

No

Are the interpretations and conclusions justified by the results?

No

Is the language acceptable?

Yes

Is it clear how to access all supporting data?

No

Do you have any ethical concerns with this paper?

No

Have you any concerns about statistical analyses in this paper?

Yes

Recommendation?

Major revision is needed (please make suggestions in comments)

Comments to the Author(s)

I have reviewed the manuscript *Man's Best Friends: Attitudes Towards the Use of Different Kinds of Animals Depend on Belief in Different Species Mental Capacities and Purpose of Use* (RSOS-181697). This paper presents a new methodological tool (the Animal Purpose Questionnaire) designed for assessing variation in the perceived acceptability of use of animals across different species and purposes. The authors have used the APQ in a survey of attitudes towards animals, finding that perceived acceptability of animal use is inversely correlated with degree of belief in animal sentience (assayed by the Belief in Animal Mind scale). They also find that female respondents and those who do not eat meat consider animal use less acceptable, and that perceived acceptability is highly variable between species and categories of use.

The APQ should prove to be a valuable instrument for measuring human attitudes to animal use across diverse species and contexts, and clearly represents an improvement over prior scales. There are many potential interesting and important uses for the scale, particularly considering current environmental crises. The survey appears well designed and results are based on a fairly large and diverse sample size. The paper is generally well written and the findings are interesting and should prompt fruitful avenues of further research. The study is particularly impressive considering the early career stage of the first two authors (Masters students).

I do however have some major concerns about the statistical analyses. None of these undermine the utility of the APQ instrument or the potential value of the paper, but do raise doubts about the findings as currently presented. Therefore these need to be resolved before the manuscript could be published in RSOS.

First of all, the authors need to justify the use of parametric statistical tests for Likert-scale data. Likert-scale data is not truly continuous and is often non-normally distributed, but the authors have not mentioned whether statistical assumptions were met. Unfortunately I could not confirm this for myself as the supplementary information does not seem to be available. The authors need to either confirm that data meet parametric assumptions, use non-parametric alternatives or use a different modelling approach altogether (e.g. ordinal regression).

Secondly, the authors should reconsider their strategy concerning multiple comparisons. The study reports a large number of statistical results, Bonferroni-corrected to account for increased risk of false-positives. This approach complicates interpretation considerably and raises questions about why so many contrasts are included. With regards to the ANOVAs, if hypotheses are non-directional (e.g. APQ scores varying between species) there seems no need for post-hoc tests and so these should simply be left out. In contrast, where directional predictions are made (e.g. APQ scores decreasing with phylogenetic closeness to humans) the post-hoc tests should be limited to

just those planned contrasts that relate to directional predictions. Alternatively, if the aim is to identify clusters of species with similar perceived acceptability, cluster analysis could be used to test this prediction more succinctly. On a related note, I think the aims of the study could be clarified in the abstract as the variation between species and purposes in APQ (seemingly one of the key findings) is not mentioned.

Finally, the authors should reconsider their analyses of the role of demographic factors in APQ scores. It is not currently clear whether the gender difference they find was predicted in advance, or on what theoretical basis this prediction is made. If the investigation of demographic factors is exploratory rather than hypothesis-driven then that is fine, it just needs to be clarified. Further, the authors should discuss the role (or lack) of religious and ethnic identity in APQ scores as this has not been fully interpreted with regards. This is particularly important given the mention of potential cultural variation (e.g. taboos surrounding pork) in APQ in the Discussion. Further, the use of Pearson's correlations to investigate relationships between APQ/BAM and demographic variables is not appropriate given that all but age are categorical variables.

Minor issues

- Meaning of '104+m' re: the minimum sample size needs explanation and citation.
- The abstract should clarify the direction of findings regarding APQ & BAM and mention the findings regarding variation in APQ/BAM across species/purposes.
- Clarify what is meant by 'comparative ratings' (e.g. page 4). Does this refer to the within-subjects design? Or presentation of rating scales for different animals simultaneously?
- Unclear what is meant by 'common-sense' (page 4)
- Use neutral terms instead of phylogenetic 'status' or 'scale' (e.g. 'position', 'placement'), unless it is clear you are referring to perceptions influenced by a 'scala naturae' schema.
- 'Species' in the title needs a possessive plural apostrophe.

Decision letter (RSOS-181697.R0)

07-May-2019

Dear Professor Cassaday:

Manuscript ID RSOS-181697 entitled "Man's Best Friends: Attitudes Towards the Use of Different Kinds of Animals Depend on Belief in Different Species Mental Capacities and Purpose of Use" which you submitted to Royal Society Open Science, has been reviewed. The comments from reviewers are included at the bottom of this letter.

In view of the criticisms of the reviewers, the manuscript has been rejected in its current form. However, a new manuscript may be submitted which takes into consideration these comments.

Please note that resubmitting your manuscript does not guarantee eventual acceptance, and that your resubmission will be subject to peer review before a decision is made.

Your resubmitted manuscript should be submitted by 04-Nov-2019. If you are unable to submit by this date please contact the Editorial Office.

on behalf of Dr Geoffrey Bird (Associate Editor) and Antonia Hamilton (Subject Editor)
openscience@royalsociety.org

Associate Editor Comments to Author (Dr Geoffrey Bird):

Associate Editor: 1

Comments to the Author:

The comments from the reviewers indicate that your manuscript requires work to be closer to a publishable standard; however, a revision would offer too little time for you to tackle the quite extensive feedback provided. With this in mind, we cannot accept the current version of the paper but would be willing to consider a resubmission that takes the concerns of the reviewers into serious consideration - good luck!

Reviewers' Comments to Author:

Reviewer: 1

Comments to the Author(s)

This is an interesting paper that is trying to achieve a reasonable goal - to identify more specifically the intersectionality of species and purpose in terms of attitudes towards animals. It is indeed true that people's attitudes towards animals, and how they should be used, varied widely. That said, it is also this variation and idiosyncratic approach to how people think about animals, which makes the goal of establishing a measure to capture this variance a somewhat difficult goal. For one thing, it suggests that such a measure should be endlessly complex (incorporating an exhaustive list of animals or purposes) if it is to map the territory. It was perhaps this question about the current paper - what the purpose of this measure is supposed to be, and what variance it is exactly measuring - which remained unclear for me and therefore reduced my enthusiasm. A few more specific points below:

1. The selection of uses seem somewhat arbitrary to me. It was unclear why medical research and basic science research should be separated out, yet use for clothes and fighting and hunting collapsed into the category 'other'.
2. The selection of animals was explained, but again my question here is what is the goal of this measure? It seemed that, in the end, it was to show that variation exists, rather than to try and pinpoint any specific attitudes about specific species. But then, I wonder what one would do with this measure, except to show that variation exists? For instance, I was very unsure what to make of the finding that there was low level agreement on purpose of use for Dolphins and Dogs? Why is this and what does it tell us?
3. The authors use the BAM to predict attitudes towards uses of different species of animals, but also acknowledge that much of the variance here may be to do with social norms - such as which animals we eat. For instance, attitudes towards eating rats is unlikely primarily determined by their perceived mental capacities. This immediately throws up the problem that the APQ is capturing all sorts of variance and this varies as a function of species and purpose, making it unclear what the instrument is measuring.

4. Some of the analyses focused on collapsing the APQ, which lead me to wonder how these analyses related to the species-specific ones.

In the end, I do think that the general aim of the paper is worthy, but I just could not clearly see what to take away from the data, or how this measure would be useful. To summarise my take, it seemed like the measure revealed that there is disagreement that animals should be used for any purpose, that there is more disagreement about some animals than others, and the belief in animal mind appears to explain whether it is considered okay in general to use an animal for a purpose. I did not find the collapsed analysis insightful, or showing anything unique to the APQ, given it was designed to investigate differences between animal and purpose, rather than collapse across them. Perhaps I have missed something, and if so then perhaps the authors could consider how they might bring out the unique contribution and purpose of this measure a little more in their explanation and treatment of it.

Reviewer: 2

Comments to the Author(s)

I have reviewed the manuscript *Man's Best Friends: Attitudes Towards the Use of Different Kinds of Animals Depend on Belief in Different Species Mental Capacities and Purpose of Use* (RSOS-181697). This paper presents a new methodological tool (the Animal Purpose Questionnaire) designed for assessing variation in the perceived acceptability of use of animals across different species and purposes. The authors have used the APQ in a survey of attitudes towards animals, finding that perceived acceptability of animal use is inversely correlated with degree of belief in animal sentience (assayed by the Belief in Animal Mind scale). They also find that female respondents and those who do not eat meat consider animal use less acceptable, and that perceived acceptability is highly variable between species and categories of use.

The APQ should prove to be a valuable instrument for measuring human attitudes to animal use across diverse species and contexts, and clearly represents an improvement over prior scales. There are many potential interesting and important uses for the scale, particularly considering current environmental crises. The survey appears well designed and results are based on a fairly large and diverse sample size. The paper is generally well written and the findings are interesting and should prompt fruitful avenues of further research. The study is particularly impressive considering the early career stage of the first two authors (Masters students).

I do however have some major concerns about the statistical analyses. None of these undermine the utility of the APQ instrument or the potential value of the paper, but do raise doubts about the findings as currently presented. Therefore these need to be resolved before the manuscript could be published in RSOS.

First of all, the authors need to justify the use of parametric statistical tests for Likert-scale data. Likert-scale data is not truly continuous and is often non-normally distributed, but the authors have not mentioned whether statistical assumptions were met. Unfortunately I could not confirm this for myself as the supplementary information does not seem to be available. The authors need to either confirm that data meet parametric assumptions, use non-parametric alternatives or use a different modelling approach altogether (e.g. ordinal regression).

Secondly, the authors should reconsider their strategy concerning multiple comparisons. The study reports a large number of statistical results, Bonferroni-corrected to account for increased risk of false-positives. This approach complicates interpretation considerably and raises questions about why so many contrasts are included. With regards to the ANOVAs, if hypotheses are non-directional (e.g. APQ scores varying between species) there seems no need for post-hoc tests and so these should simply be left out. In contrast, where directional predictions are made (e.g. APQ scores decreasing with phylogenetic closeness to humans) the post-hoc tests should be limited to just those planned contrasts that relate to directional predictions. Alternatively, if the aim is to

identify clusters of species with similar perceived acceptability, cluster analysis could be used to test this prediction more succinctly. On a related note, I think the aims of the study could be clarified in the abstract as the variation between species and purposes in APQ (seemingly one of the key findings) is not mentioned.

Finally, the authors should reconsider their analyses of the role of demographic factors in APQ scores. It is not currently clear whether the gender difference they find was predicted in advance, or on what theoretical basis this prediction is made. If the investigation of demographic factors is exploratory rather than hypothesis-driven then that is fine, it just needs to be clarified. Further, the authors should discuss the role (or lack) of religious and ethnic identity in APQ scores as this has not been fully interpreted with regards. This is particularly important given the mention of potential cultural variation (e.g. taboos surrounding pork) in APQ in the Discussion. Further, the use of Pearson's correlations to investigate relationships between APQ/BAM and demographic variables is not appropriate given that all but age are categorical variables.

Minor issues

- Meaning of '104+m' re: the minimum sample size needs explanation and citation.
- The abstract should clarify the direction of findings regarding APQ & BAM and mention the findings regarding variation in APQ/BAM across species/purposes.
- Clarify what is meant by 'comparative ratings' (e.g. page 4). Does this refer to the within-subjects design? Or presentation of rating scales for different animals simultaneously?
- Unclear what is meant by 'common-sense' (page 4)
- Use neutral terms instead of phylogenetic 'status' or 'scale' (e.g. 'position', 'placement'), unless it is clear you are referring to perceptions influenced by a 'scala naturae' schema.
- 'Species' in the title needs a possessive plural apostrophe.

Author's Response to Decision Letter for (RSOS-181697.R0)

See Appendix A.

RSOS-191162.R0

Review form: Reviewer 1

Is the manuscript scientifically sound in its present form?

Yes

Are the interpretations and conclusions justified by the results?

Yes

Is the language acceptable?

Yes

Do you have any ethical concerns with this paper?

No

Have you any concerns about statistical analyses in this paper?

No

Recommendation?

Accept with minor revision (please list in comments)

Comments to the Author(s)

I appreciated the authors clarification of my initial reaction to the manuscript. While I agreed with the general approach and content, I found the measure (and its purpose) hard to understand. I feel this has been made much clearer now. Perhaps one thing, however, is that the authors could frame this more as a demonstration measure that could be adapted for different purposes/comparisons. I don't think that anyone would want to try and come up with an endless list of animals or purposes, but perhaps this measure is more like a blueprint concept that people can draw on to make more specific and focused comparisons between how people think about the use of different animals for different purposes. I feel that this would be a better representation of this measure, which is not undergone any rigorous psychometric testing etc. For me the contribution is showing that people distinguish between animals and how they are used, rather than the contribution of a specific tool that can be meaningfully adopted elsewhere. I would encourage the authors to consider this somewhat minor (although important) reframing of their paper and its contribution.

Beyond this, I noticed a few small typo's that could be addressed in the next revision.

Review form: Reviewer 2**Is the manuscript scientifically sound in its present form?**

No

Are the interpretations and conclusions justified by the results?

No

Is the language acceptable?

Yes

Do you have any ethical concerns with this paper?

No

Have you any concerns about statistical analyses in this paper?

Yes

Recommendation?

Major revision is needed (please make suggestions in comments)

Comments to the Author(s)

I have reviewed the revised manuscript: "Man's Best Friends: Attitudes Towards the Use of Different Kinds of Animal Depend on Belief in Different Species' Mental Capacities and Purpose of Use" (RSOS-191162). While I think the study has merit and I appreciate the effort the authors have gone to to respond to the Reviewers' comments, substantial issues with the statistical analysis remain, sufficient to preclude publication in RSOS at present. These issues are as follows:

Modelling Likert-scales as continuous variables

Despite what the authors state in the responses and revised manuscript, Likert scales are ordinal, not interval scales (p. 13). Although they have a rank order, we cannot necessarily assume that differences between points on the scale are equivalent. I do appreciate that parametric statistics

may produce similar results to non-parametric tests when samples are large and data are approximately normally distributed, but to make a really convincing case the authors would need to be able to actually show that results do not substantially differ when using alternative approaches. The authors could at least repeat one or two of their main analyses using more appropriate methods to see if anything changes. The authors mention that non-parametric alternatives to mixed-ANOVA and multiple regression are not reliable, but they have not explained why they do not use ordinal regression as suggested in my original review. While Likert-scales are commonly treated as continuous in psychological research, this can cause serious problems including false positives, false negatives and reversed signs (Liddell and Kruschke, 2018). I appreciate that ordinal regression models can be challenging to implement, but they are becoming more commonly used (see e.g. (Brand and Mesoudi, 2019)). Moreover, tutorial materials, including R-scripts, are now available (see e.g. (Liddell and Kruschke, 2018; Bürkner and Vuorre, 2019)).

Hypothesis testing

The study currently uses a combination of exploratory and hypothesis-driven approaches. For example, directional predictions are tested regarding the effect of phylogenetic position on attitudes towards animals (p. 7), while predictions are not made for the effects of all demographic variables. I appreciate that the current study is challenging because some predictions are difficult to specify a priori, but the authors need to explain and justify their approach much more explicitly, stating which analyses are exploratory and which driven by prior predictions in advance. As a consequence of this mixed, partly exploratory approach, the authors end up having to report both main effects from ANOVAs as well as long lists of Bonferroni-corrected individual comparisons in the results section, the importance of which is often unclear. As previously suggested (and demonstrated by the authors in their response, but not implemented in the paper), a clustering approach would be a far more efficient means of exploratory analysis of variation in attitudes to animals by species and purpose.

Similarly, the authors need to much more explicitly clarify predictions in relation to demographic variables. Although directional predictions are made only in relation to gender and diet, the authors cite several studies in the Introduction showing effects of other demographic variables (e.g. age, religion, culture) on attitudes towards animal use (p. 4). It is not currently clear why directional predictions could not be made for these variables as well.

Collapsing scores

In order to explore the interaction between species and purpose as drivers of attitudes towards animal use, the authors collapse scores across species to compare purpose categories, and collapse scores across purpose to compare species categories (p. 12). This is unnecessary and results in a loss of statistical power; multi-level models could be used instead to avoid this and maintain greater power.

Minor issues

- Why is it necessary to use t-tests to calculate significance values for the effects of predictors from multiple regression analyses? (p. 12/13)
- It would be helpful to share the SPSS output file as well as the raw data with Reviewers so we can confirm details of the analysis, data processing, etc.
- The authors do not explain what sphericity is or why it is important to check (p. 13)
- Predictions regarding the effects of phylogenetic position should still hold despite other factors, if the effect is strong enough. If the authors wish to make specific predictions regarding moderators of this effect, these could be tested explicitly in the multiple regression models (p. 7)
- The rule of thumb used for sample sizes is not sufficiently justified – the number 104 seems completely arbitrary (p. 15).
- Phylogenetic ‘scale’ should be avoided in favour of a neutral term on p. 19

Bibliography

- Brand, C. O. and Mesoudi, A. (2019) 'Prestige and dominance-based hierarchies exist in naturally occurring human groups, but are unrelated to task-specific knowledge', *Royal Society Open Science*. The Royal Society, 6(5), p. 181621. doi: 10.1098/rsos.181621.
- Bürkner, P.-C. and Vuorre, M. (2019) 'Ordinal Regression Models in Psychology: A Tutorial', *Advances in Methods and Practices in Psychological Science*. SAGE PublicationsSage CA: Los Angeles, CA, 2(1), pp. 77–101. doi: 10.1177/2515245918823199.
- Liddell, T. M. and Kruschke, J. K. (2018) 'Analyzing ordinal data with metric models: What could possibly go wrong?', *Journal of Experimental Social Psychology*. Academic Press, 79, pp. 328–348. doi: 10.1016/J.JESP.2018.08.009.

Decision letter (RSOS-191162.R0)

24-Sep-2019

Dear Professor Cassaday,

The Subject Editor assigned to your paper ("Man's Best Friends: Attitudes Towards the Use of Different Kinds of Animal Depend on Belief in Different Species' Mental Capacities and Purpose of Use") has now received comments from reviewers. We would like you to revise your paper in accordance with the referee and Associate Editor suggestions which can be found below (not including confidential reports to the Editor). Please note this decision does not guarantee eventual acceptance.

Please submit a copy of your revised paper before 17-Oct-2019. Please note that the revision deadline will expire at 00.00am on this date. If we do not hear from you within this time then it will be assumed that the paper has been withdrawn. In exceptional circumstances, extensions may be possible if agreed with the Editorial Office in advance. We do not allow multiple rounds of revision so we urge you to make every effort to fully address all of the comments at this stage. If deemed necessary by the Editors, your manuscript will be sent back to one or more of the original reviewers for assessment. If the original reviewers are not available we may invite new reviewers.

When submitting your revised manuscript, you must respond to the comments made by the referees and upload a file "Response to Referees" in "Section 6 - File Upload". Please use this to document how you have responded to each of the comments, and the adjustments you have made. In order to expedite the processing of the revised manuscript, please be as specific as possible in your response.

- Ethics statement

If your study uses humans or animals please include details of the ethical approval received, including the name of the committee that granted approval. For human studies please also detail

whether informed consent was obtained. For field studies on animals please include details of all permissions, licences and/or approvals granted to carry out the fieldwork.

- Data accessibility

If you wish to submit your supporting data or code to Dryad (<http://datadryad.org/>), or modify your current submission to dryad, please use the following link:
<http://datadryad.org/submit?journalID=RSOS&manu=RSOS-191162>

- Competing interests

- Authors' contributions

- Acknowledgements

- Funding statement

Kind regards,

Andrew Dunn

on behalf of Dr Geoffrey Bird (Associate Editor) and Antonia Hamilton (Subject Editor)
 openscience@royalsociety.org

Subject Editor comments

- I have looked into the stats issues raised by Rev 2 carefully. You do not need to remodel your data with Likert scales as ordinal as long as you can show it fits a normal distribution. You also do not need to address every one of the minor stats comments, as many of these issues are standard in psychology. It would be useful to do multi-level modelling if you can.

Reviewer comments to Author:

Reviewer: 1

Comments to the Author(s)

I appreciated the authors clarification of my initial reaction to the manuscript. While I agreed with the general approach and content, I found the measure (and its purpose) hard to understand. I feel this has been made much clearer now. Perhaps one thing, however, is that the authors could frame this more as a demonstration measure that could be adapted for different purposes/comparisons. I don't think that anyone would want to try and come up with an endless list of animals or purposes, but perhaps this measure is more like a blueprint concept that people can draw on to make more specific and focused comparisons between how people think about the use of different animals for different purposes. I feel that this would be a better representation of this measure, which is not undergone any rigorous psychometric testing etc. For me the contribution is showing that people distinguish between animals and how they are used, rather than the contribution of a specific tool that can be meaningfully adopted elsewhere. I would encourage the authors to consider this somewhat minor (although important) reframing of their paper and its contribution.

Beyond this, I noticed a few small typo's that could be addressed in the next revision.

Reviewer: 2

Comments to the Author(s)

I have reviewed the revised manuscript: "Man's Best Friends: Attitudes Towards the Use of Different Kinds of Animal Depend on Belief in Different Species' Mental Capacities and Purpose of Use" (RSOS-191162). While I think the study has merit and I appreciate the effort the authors have gone to to respond to the Reviewers' comments, substantial issues with the statistical analysis remain, sufficient to preclude publication in RSOS at present. These issues are as follows:

Modelling Likert-scales as continuous variables

Despite what the authors state in the responses and revised manuscript, Likert scales are ordinal, not interval scales (p. 13). Although they have a rank order, we cannot necessarily assume that differences between points on the scale are equivalent. I do appreciate that parametric statistics may produce similar results to non-parametric tests when samples are large and data are approximately normally distributed, but to make a really convincing case the authors would need to be able to actually show that results do not substantially differ when using alternative approaches. The authors could at least repeat one or two of their main analyses using more appropriate methods to see if anything changes. The authors mention that non-parametric alternatives to mixed-ANOVA and multiple regression are not reliable, but they have not explained why they do not use ordinal regression as suggested in my original review. While Likert-scales are commonly treated as continuous in psychological research, this can cause

serious problems including false positives, false negatives and reversed signs (Liddell and Kruschke, 2018). I appreciate that ordinal regression models can be challenging to implement, but they are becoming more commonly used (see e.g. (Brand and Mesoudi, 2019)). Moreover, tutorial materials, including R-scripts, are now available (see e.g. (Liddell and Kruschke, 2018; Bürkner and Vuorre, 2019)).

Hypothesis testing

The study currently uses a combination of exploratory and hypothesis-driven approaches. For example, directional predictions are tested regarding the effect of phylogenetic position on attitudes towards animals (p. 7), while predictions are not made for the effects of all demographic variables. I appreciate that the current study is challenging because some predictions are difficult to specify a priori, but the authors need to explain and justify their approach much more explicitly, stating which analyses are exploratory and which driven by prior predictions in advance. As a consequence of this mixed, partly exploratory approach, the authors end up having to report both main effects from ANOVAs as well as long lists of Bonferroni-corrected individual comparisons in the results section, the importance of which is often unclear. As previously suggested (and demonstrated by the authors in their response, but not implemented in the paper), a clustering approach would be a far more efficient means of exploratory analysis of variation in attitudes to animals by species and purpose.

Similarly, the authors need to much more explicitly clarify predictions in relation to demographic variables. Although directional predictions are made only in relation to gender and diet, the authors cite several studies in the Introduction showing effects of other demographic variables (e.g. age, religion, culture) on attitudes towards animal use (p. 4). It is not currently clear why directional predictions could not be made for these variables as well.

Collapsing scores

In order to explore the interaction between species and purpose as drivers of attitudes towards animal use, the authors collapse scores across species to compare purpose categories, and collapse scores across purpose to compare species categories (p. 12). This is unnecessary and results in a loss of statistical power; multi-level models could be used instead to avoid this and maintain greater power.

Minor issues

- Why is it necessary to use t-tests to calculate significance values for the effects of predictors from multiple regression analyses? (p. 12/13)
- It would be helpful to share the SPSS output file as well as the raw data with Reviewers so we can confirm details of the analysis, data processing, etc.
- The authors do not explain what sphericity is or why it is important to check (p. 13)
- Predictions regarding the effects of phylogenetic position should still hold despite other factors, if the effect is strong enough. If the authors wish to make specific predictions regarding moderators of this effect, these could be tested explicitly in the multiple regression models (p. 7)
- The rule of thumb used for sample sizes is not sufficiently justified – the number 104 seems completely arbitrary (p. 15).
- Phylogenetic ‘scale’ should be avoided in favour of a neutral term on p. 19

Bibliography

Brand, C. O. and Mesoudi, A. (2019) ‘Prestige and dominance-based hierarchies exist in naturally occurring human groups, but are unrelated to task-specific knowledge’, *Royal Society Open Science*. The Royal Society, 6(5), p. 181621. doi: 10.1098/rsos.181621.

Bürkner, P.-C. and Vuorre, M. (2019) 'Ordinal Regression Models in Psychology: A Tutorial', *Advances in Methods and Practices in Psychological Science*. SAGE PublicationsSage CA: Los Angeles, CA, 2(1), pp. 77–101. doi: 10.1177/2515245918823199.

Liddell, T. M. and Kruschke, J. K. (2018) 'Analyzing ordinal data with metric models: What could possibly go wrong?', *Journal of Experimental Social Psychology*. Academic Press, 79, pp. 328–348. doi: 10.1016/J.JESP.2018.08.009.

Author's Response to Decision Letter for (RSOS-191162.R0)

See Appendix B.

Decision letter (RSOS-191162.R1)

21-Jan-2020

Dear Professor Cassaday,

It is a pleasure to accept your manuscript entitled "Man's Best Friends: Attitudes Towards the Use of Different Kinds of Animal Depend on Belief in Different Species' Mental Capacities and Purpose of Use" in its current form for publication in *Royal Society Open Science*. The comments of the reviewer(s) who reviewed your manuscript are included at the foot of this letter.

Thank you for your fine contribution. On behalf of the Editors of *Royal Society Open Science*, we look forward to your continued contributions to the Journal.

Kind regards,
Anita Kristiansen
Editorial Coordinator
Royal Society Open Science
openscience@royalsociety.org

on behalf of Dr Geoffrey Bird (Associate Editor) and Antonia Hamilton (Subject Editor)
openscience@royalsociety.org

Appendix A

Reviewer: 1

Comments to the Author(s)

This is an interesting paper that is trying to achieve a reasonable goal – to identify more specifically the intersectionality of species and purpose in terms of attitudes towards animals. It is indeed true that people's attitudes towards animals, and how they should be used, varied widely. That said, it is also this variation and idiosyncratic approach to how people think about animals, which makes the goal of establishing a measure to capture this variance a somewhat difficult goal. For one thing, it suggests that such a measure should be endlessly complex (incorporating an exhaustive list of animals or purposes) if it is to map the territory. It was perhaps this question about the current paper – what the purpose of this measure is supposed to be, and what variance it is exactly measuring – which remained unclear for me and therefore reduced my enthusiasm.

The intention is that the approach should be used flexibly rather than to exhaustively sample a very long list of species and purposes. For example, the comparison between the purposes of basic science and medical research would be of particular interest following sessions intended to promote public understanding of animal research. We see food production and pest control as benchmark purposes of use. We agree that the category of 'other' is somewhat simplistic but would see the present study as providing a useful template for future work to address further distinctions of interest. There was some discussion of how the APQ might be used in future studies (p21, 23) but the study objectives were not clearly set out in the Introduction and this has been remedied p6-7. As Reviewer 2 points out, the development of such measures is timely given the potentially devastating impact of current environmental crises on biodiversity.

Specific points

1. The selection of uses seem somewhat arbitrary to me. It was unclear why medical research and basic science research should be separated out, yet use for clothes and fighting and hunting collapsed into the category 'other'.

We agree that the selection of uses could be seen as arbitrary depending on the interests of the reader. However, the distinction between using animals for medical research (with some more immediate likely health benefit for humans, as well as potentially other animals) versus basic science research (with no immediate likely health benefit) is an important one, both from the point of view of the legislation and well as from the perspective of interested publics. For example, approvals of Project Licences by the UK Home Office require the completion of a cost-benefit analysis. Vegetarians might see medical research as a more essential use of animals than eating them; appreciation of the importance of basic science research might depend on level of education (measured in the present study but not examined in the sense that the majority of the sample was highly educated as shown in Table 1).

The selection of uses has been clarified in the Introduction p6 and we have added consideration of the (lack of) distinction between basic science and medical research to the Discussion p18. The adaptation of the APQ template to explore additional purposes of use has been explained p23. We have also included consideration of the possibility that (in the present study) relatively higher levels of disagreement with the use of animals for 'other' purposes could relate to the ambiguity and/or 'catch all' nature of this category.

2. The selection of animals was explained, but again my question here is what is the goal of this measure? It seemed that, in the end, it was to show that variation exists, rather than to try and pinpoint any specific attitudes about specific species. But then, I wonder what one would do with this measure, except to show that variation exists? For instance, I was very unsure what to make of the finding that there was low level agreement on purpose of use for Dolphins and Dogs? Why is this and what does it tell us?

We're interested in statistically significant systematic differences in levels of concern by species and purpose and how such differences relate to belief in animal mind and demographics such as gender and dietary preferences (as well as other factors such as culture and religious beliefs in the longer term). The relatively greater concern for dolphins and dogs over chimpanzees for example is of interest because primates might be expected to receive the greatest moral consideration (Herzog & Galvin, 1997; Intro p5). The role of perceived behavioural similarity is further discussed p22.

As discussed (p23), there is also scope to use the APQ template as a tool to test the effectiveness of interventions intended for example to promote understanding of animal research.

3. The authors use the BAM to predict attitudes towards uses of different species of animals, but also acknowledge that much of the variance here may be to do with social norms – such as which animals we eat. For instance, attitudes towards eating rats is unlikely primarily determined by their perceived mental capacities. This immediately throws up the problem that the APQ is capturing all sorts of variance and this varies as a function of species and purpose, making it unclear what the instrument is measuring.

We agree that interpretation of the species by purpose interaction (in the factorial design of the APQ) must take account of social norms, particularly in connection with animals which are rarely eaten and may be perceived as disgusting in the context of food production (at least in the dominant cultures of the UK, as discussed p20).

Nonetheless, systematic data on differences in attitude by species and purpose can still be of interest. Animal welfare activists actively campaign against the use of laboratory rodents, yet their routine extermination in the course of pest control attracts relatively little attention. The APQ provides a tool to measure how this practical distinction relates to reported attitudes. We've improved the introduction to the use of the APQ (p6-7) and elaborated on the discussion of the findings (p21, 23).

4. Some of the analyses focused on collapsing the APQ, which lead me to wonder how these analyses related to the species-specific ones.

Particularly given that attitudes towards eating different animals may be a large driver of the species x purpose interaction, for the regression analyses it was appropriate to collapse across species (to test how demographic factors might account for attitudes in relation to purpose) or purpose (to test how demographic factors might account for attitudes in relation to species). The presentation of these further analyses has been improved p12 and they have been further discussed p18-19, 21-22.

In the end, I do think that the general aim of the paper is worthy, but I just could not clearly see what to take away from the data, or how this measure would be useful. To summarise my take, it seemed like the measure revealed that there is disagreement that animals should be used for any purpose, that there is more disagreement about some animals than others, and the belief in animal mind appears to explain whether it is considered okay in general to use an animal for a purpose.

Yes, the manuscript presents a method to quantify and determine the significance of differences in attitudes to the use of animals. A number of the differences identified will be in line with readers' lay expectations, (depending on their culture) others may not be. The importance of systematic evidence on this point has been given greater emphasis in the revised manuscript (e.g., in the Intro p6-7).

I did not find the collapsed analysis insightful, or showing anything unique to the APQ, given it was designed to investigate differences between animal and purpose, rather than collapse across them.

Systematic evidence on attitudes by purpose is also lacking. The APQ gives us evidence by both species (to test how demographic factors might account for attitudes in relation to purpose) and purpose (to test how demographic factors might account for attitudes in relation to species). As suggested, in the revision, we have improved the presentation (p12 and elsewhere), to bring out the

unique contribution of the measure.

Reviewer: 2

Comments to the Author(s)

I have reviewed the manuscript **Man's Best Friends: Attitudes Towards the Use of Different Kinds of Animals Depend on Belief in Different Species Mental Capacities and Purpose of Use (RSOS-181697)**. This paper presents a new methodological tool (the Animal Purpose Questionnaire) designed for assessing variation in the perceived acceptability of use of animals across different species and purposes. The authors have used the APQ in a survey of attitudes towards animals, finding that perceived acceptability of animal use is inversely correlated with degree of belief in animal sentience (assayed by the Belief in Animal Mind scale). They also find that female respondents and those who do not eat meat consider animal use less acceptable, and that perceived acceptability is highly variable between species and categories of use.

The reviewer has provided a fine summary of the main findings of the study.

The APQ should prove to be a valuable instrument for measuring human attitudes to animal use across diverse species and contexts, and clearly represents an improvement over prior scales. There are many potential interesting and important uses for the scale, particularly considering current environmental crises. The survey appears well designed and results are based on a fairly large and diverse sample size. The paper is generally well written and the findings are interesting and should prompt fruitful avenues of further research. The study is particularly impressive considering the early career stage of the first two authors (Masters students).

The reviewer appreciates the point of study and we have provided further clarification in response to Reviewer 1.

I do however have some major concerns about the statistical analyses. None of these undermine the utility of the APQ instrument or the potential value of the paper, but do raise doubts about the findings as currently presented. Therefore these need to be resolved before the manuscript could be published in RSOS.

First of all, the authors need to justify the use of parametric statistical tests for Likert-scale data. Likert-scale data is not truly continuous and is often non-normally distributed, but the authors have not mentioned whether statistical assumptions were met. Unfortunately I could not confirm this for myself as the supplementary information does not seem to be available. The authors need to either confirm that data meet parametric assumptions, use non-parametric alternatives or use a different modelling approach altogether (e.g. ordinal regression).

The raw Qualtrics output was submitted with the manuscript and will be deposited in the University of Nottingham Research Data Repository and made freely available post-publication, along with the SPSS file.

The survey scales used bipolar adjectives, hence participants in effect rated semantic differential on an interval scale measuring levels of (dis-)agreement and parametric tests are in principle applicable (there were 5 levels to the AAS and APQ scales and 7 levels of the BAM). There remain widely held objections to the parametric analysis of Likert-derived data but sample size and normality of the distribution can provide some justification for a parametric approach (Jamieson, 2004). Parametric tests are more sensitive and vastly more powerful than their non-parametric alternatives (there are no good non-parametric alternatives for a mixed-ANOVA or a multiple regression). Parametric tests are also very robust, so even when their assumptions are violated there is low risk of drawing the incorrect conclusion (Norman, 2010). Both AAS and BAM (total scores) have been analysed parametrically in a number of previous studies.

In the present study, we had relatively large N. Parametric tests assume normality of the distribution of the means and Central Limit Theorem shows that (even for sample sizes very much lower than the one used in the present study) the means will approximate a normal distribution, irrespective of distributions of the raw data (Norman, 2010). Moreover, the data were totalled over in this case 317 Likert scales and (in some cases) over a number of items (e.g. in consideration of main effects for the APQ). Different considerations apply to correlational and regression approaches which are more sensitive to individual data at the extremes of the distribution but which have nonetheless been shown to be robust in simulation studies (Havlicek & Petersen, 1976). In the present study, the data approximated a normal distribution for both APQ and BAM scores and the SPSS output is provided (for the reviewer to see) below. In the revision, justification of the parametric approaches adopted has been provided p13.

We have checked and we had reported the Greenhouse-Geisser corrected values because the sphericity assumption was violated for the main effects of species and purpose and for the species x purpose interaction. In the revision (p13), we have included a sentence to confirm that Mauchly's test of sphericity indicated that the assumption of sphericity had been violated for Species, Purpose and Species x Purpose, hence Greenhouse-Geisser corrected values have been reported.

Secondly, the authors should reconsider their strategy concerning multiple comparisons. The study reports a large number of statistical results, Bonferroni-corrected to account for increased risk of false-positives. This approach complicates interpretation considerably and raises questions about why so many contrasts are included. With regards to the ANOVAs, if hypotheses are non-directional (e.g. APQ scores varying between species) there seems no need for post-hoc tests and so these should simply be left out. In contrast, where directional

predictions are made (e.g. APQ scores decreasing with phylogenetic closeness to humans) the post-hoc tests should be limited to just those planned contrasts that relate to directional predictions. Alternatively, if the aim is to identify clusters of species with similar perceived acceptability, cluster analysis could be used to test this prediction more succinctly. On a related note, I think the aims of the study could be clarified in the abstract as the variation between species and purposes in APQ (seemingly one of the key findings) is not mentioned.

We believe that there is value in knowing in which cases attitude ratings differ significantly by species. We take the point that more focused (planned) comparisons could be justified but prefer a more conservative (Bonferroni corrected) approach because of the difficulty in specifying which differences should be significant *a priori*. Phylogenetic closeness is just one of a number of factors likely to influence participants' attitudes with respect to particular species. Other likely factors include the extent to which the species has been domesticated (and hence relatively safe to pet) and the perceived attractiveness ('cuteness') of the species. Indeed the Bonferroni corrected findings bear out the proposition that phylogenetic closeness is not the only driver in that Chimpanzee was not given the lowest rating (reflecting highest levels of disagreement with use).

Similarly, we could not assume which differences by purpose should be significant *a priori* and whilst we had some expectations, precisely because evidence on this point has hitherto been lacking, there were insufficient grounds to make clear predictions. For example one outcome of interest which we had no basis to predict was the lack of any significant difference between levels of support for animal use in Medical Research and Basic Science Research in the population sampled (whereas all other differences between the purposes of use survived Bonferroni correction).

As suggested, the aims of the study have been clarified in the Abstract (as well as in the Intro), to include examination of the variation by species and purposes examined using the APQ. We had not aimed to identify clusters of species with similar perceived acceptability of use *a priori* and had not intended to report clusters of animals as real constructs (characterised by high within-group and low between-group homogeneity; Clatworthy et al., 2005). In presentation of the findings we intended to highlight similarly rated species, to help to contextualise the ratings of the individual species under consideration (which were in some cases higher than might have been expected based on phylogeny). We have nonetheless conducted some cluster analysis (using SPSS) in response to Reviewer 2's suggestion and the dendrogram output (using average linkage between groups) is shown below for (A) the BAM and (B) the APQ scores.

(A)

(B)

The cluster analyses confirm a cluster for the APQ ratings (dog, dolphin, chimp together) but there is little differentiation between the other 9 animal species. The BAM output is more differentiated showing three distinct clusters but we feel such an analysis focused on similarity (as the obverse of difference between species) does not add to the manuscript. Moreover, we do not wish to suggest a classification of animals (of which we sample just a few selected species) based on BAM or attitudes to their use (Clatworthy et al., 2005). In the revised manuscript, we have removed all use of the term cluster and rephrased (p14) to avoid appearing to make any such suggestion.

Finally, the authors should reconsider their analyses of the role of demographic factors in APQ scores. It is not currently clear whether the gender difference they find was predicted in advance, or on what theoretical basis this prediction is made. If the investigation of demographic factors is exploratory rather than hypothesis-driven then that is fine, it just needs to be clarified. Further, the authors should discuss the role (or lack) of religious and ethnic identity in APQ scores as this has not been fully interpreted with regards. This is particularly important given the mention of potential cultural variation (e.g. taboos surrounding pork) in APQ in the Discussion. Further, the use of Pearson's correlations to investigate relationships between APQ/BAM and demographic variables is not appropriate given that all but age are categorical variables.

The justification for the inclusion of demographic factors has been improved p7. The gender differences indicating stronger pro-welfare attitudes in females were expected based on the studies cited in the Discussion p19 (which are now also cited in the Introduction p7). However, we had no basis to predict the finding that females showed no corresponding increase in BAM.

The (lack of) identified roles for religious and ethnic identity in APQ scores has been discussed p19. As shown in Table 1 these analyses were necessarily crude – we included a wide range of response options but (whilst reasonably representative of the UK population) the sample was not very diverse.

Pearson correlations can be calculated between continuous variables and dichotomous categorical variables, as point-biserial correlations, provided there are no outliers for the continuous variable for each category of the dichotomous variable and provided the continuous variable is approximately normally distributed and with equal variances for each category of the dichotomous variable (see e.g.

<https://statistics.laerd.com/spss-tutorials/point-biserial-correlation-using-spss-statistics.php>).

Dichotomous variables can similarly be entered into multiple regression (and categorical variables dummy coded). The categorical variables such as diet were recoded as dichotomous variables ('dummy coding') as explained p12. The items coded as dummy variables are also in bold in Table 1.

Minor issues

- **Meaning of '104+m' re: the minimum sample size needs explanation and citation.**

This has been clarified and a citation provided p 15. '104+m' is a rule of thumb for the suggested minimum number of participants in relation to the number of predictors (m) for the partial correlation. More complex rules of thumb take effect size into account as well as the number of predictors (Green, 1991). Effect size was unknown in the present study but the sample sizes achieved (317, with 272 providing all of the suggested demographic information) were more than double that suggested by the algorithm. In Table 1 summarising the demographics we stated N=317 but not all participants provided all the requested demographic information, so the table legend has been corrected to read N=272-317.

- **The abstract should clarify the direction of findings regarding APQ & BAM and mention the findings regarding variation in APQ/BAM across species/purposes.**

As suggested, the abstract now makes the findings more explicit.

- **Clarify what is meant by 'comparative ratings' (e.g. page 4). Does this refer to the within-subjects design? Or presentation of rating scales for different animals simultaneously?**

The ratings were comparative in the sense that participants were required to rate a number of species, one after the other (generating within-subjects data). This has been clarified p4 and p7-8.

- **Unclear what is meant by 'common-sense' (page 4)**

This term (now also in quotation marks in the manuscript text) was used in the cited Herzog and Galvin (1997) study to describe knowledge based on direct daily experience (the term is used similarly in the animal rights literature). This has been clarified p4.

- **Use neutral terms instead of phylogenetic 'status' or 'scale' (e.g. 'position', 'placement'), unless it is clear you are referring to perceptions influenced by a 'scala naturae' schema.**

This has been corrected throughout (as suggested, references to phylogenetic status or scale have been replaced by position or placement).

- **'Species' in the title needs a possessive plural apostrophe.**

This has been corrected.

Additional References

Clatworthy, J., Buick, D., Hankins, M., Weinman, J., & Horne R. (2005). The use and reporting of cluster analysis in health psychology: A review. *British Journal of Health Psychology*, 10, 329-358.

Green S.B. (1991). How many subjects does it take to do a regression analysis. *Multivariate Behavioral Research*, 26(3), 499-510. doi: 10.1207/s15327906mbr2603_7.

Havlicek, L.L., & Peterson, N.L. (1976). Robustness of the Pearson correlation against violations of assumptions. *Perceptual and Motor Skills*, 43 (3), 1319-1334. doi: 10.2466/pms.1976.43.3f.1319

Jamieson, S. (2004). Likert scales: how to (ab)use them. *Medical Education*, 38(12), 1217-1218. doi: 10.1111/j.1365-2929.2004.02012.x

Norman, G. (2010). Likert scales, levels of measurement and the “laws” of statistics. *Advances in Health Sciences Education: Theory and Practice*, 15(5), 625-632. doi: 10.1007/s10459-010-9222-y

Appendix B

Royal Society Open Science
Journal Editor: Andrew Dunn
MS Reference Number: RSOS-191162

Subject Editor comments

- I have looked into the stats issues raised by Rev 2 carefully. You do not need to remodel your data with Likert scales as ordinal as long as you can show it fits a normal distribution. You also do not need to address every one of the minor stats comments, as many of these issues are standard in psychology. It would be useful to do multi-level modelling if you can.

We appreciate that the Subject Editor has taken the time to consider the stats issues raised by Reviewer 2. We confirm that the distribution of the data was suitable for parametric analysis. The relevant SPSS output has been provided in response to Reviewer 2, in evidence that the p13 statement to this effect in the manuscript is justified. We have further addressed the minor stats comments as far as possible, following the advice that further revision of the text is not necessary in the case of the issues that are typical in this line of research. We can't give multi-level modelling full consideration within the 3-week deadline for the revisions but have investigated this approach.

Multilevel models are particularly appropriate for research designs in which participants' data are organised at more than one level. We don't have a priori nested data in the present study, the list of species was limited and species were not selected with sub-groupings in mind. The 'pest, profit, pet' distinction was used as a guide in animal selection but not intended to provide an absolute structure to the data. Moreover, BAM wasn't predicted to vary with this distinction. So it's also not clear what the level 2 groupings should be for a multilevel modelling approach: the cluster analysis for the APQ only suggested one distinct sub-group/ 2 groupings; maximum 3 groupings in the case of the BAM ratings (please see the below cluster analysis output). We appreciate that there is no necessary limit on the number of groups required for multilevel modelling (and certainly not the 20 groups recommended in some of the earlier texts) but there can be issues with small numbers of groups, as covered for example in Andrew Gelman's blog - https://statmodeling.stat.columbia.edu/2007/08/16/no_you_dont_nee/#.XZHE6w_TYHU.mailto

We agree that future uses of the new APQ scale would benefit from picking animal groups in a nested structure for the type of multilevel analysis suggested (and/or cluster analyses, please see also below). A comment to this effect has been included in the revised Discussion, p23.

We have followed the suggestion to provide the data also in SPSS format and will also make the data available in that format in the University of Nottingham Research Data Repository post-publication. Thus the data will be freely available for all manner of secondary analyses.

Reviewer: 1

I appreciated the author's clarification of my initial reaction to the manuscript. While I agreed with the general approach and content, I found the measure (and its purpose) hard to understand. I feel this has been made much clearer now. Perhaps one thing, however, is that

the authors could frame this more as a demonstration measure that could be adapted for different purposes/comparisons. I don't think that anyone would want to try and come up with an endless list of animals or purposes, but perhaps this measure is more like a blueprint concept that people can draw on to make more specific and focused comparisons between how people think about the use of different animals for different purposes. I feel that this would be a better representation of this measure, which is not undergone any rigorous psychometric testing etc...

As suggested, we have further revised the framing of the study in the Intro, p7-8.

Beyond this, I noticed a few small typo's that could be addressed in the next revision.

We have had the manuscript read by a colleague who was not associated with the project, to provide a fresh pair of eyes. Based on her close reading of the manuscript and the authors' further checks, we have corrected all the typos we can find.

Reviewer: 2

While I think the study has merit and I appreciate the effort the authors have gone to respond to the Reviewers' comments, substantial issues with the statistical analysis remain...

The Reviewer is not fully satisfied with our original responses to statistical points raised. We appreciate that there can be divergent views on the appropriateness of statistical approaches and trust that the recommendation of the editor will be taken into account in evaluation of the below responses. Moreover, although we were hesitant to give the findings away pre-publication we'd be delighted to share the data post-publication, also in SPSS format, so interested parties can further analyse the data as they see fit.

Modelling Likert-scales as continuous variables

Despite what the authors state in the responses and revised manuscript, Likert scales are ordinal, not interval scales (p. 13). Although they have a rank order, we cannot necessarily assume that differences between points on the scale are equivalent. I do appreciate that parametric statistics may produce similar results to non-parametric tests when samples are large and data are approximately normally distributed, but to make a really convincing case the authors would need to be able to actually show that results do not substantially differ when using alternative approaches. The authors could at least repeat one or two of their main analyses using more appropriate methods to see if anything changes. The authors mention that non-parametric alternatives to mixed-ANOVA and multiple regression are not reliable, but they have not explained why they do not use ordinal regression as suggested in my original review. While Likert-scales are commonly treated as continuous in psychological research, this can cause serious problems including false positives, false negatives and reversed signs (Liddell and Kruschke, 2018)...

The editor suggests that we do not need to remodel the Likert scale data using ordinal regression if we can show it fits a normal distribution. As stated in the Results p13, the distribution of the data was suitable for parametric approaches. The SPSS output is provided (in evidence of this p13 statement) below.

We also provided some statistical justification p13, to which we have added the reference suggested by the reviewer (Liddell & Kruschke, 2018) as a 'but see...', in the interest of balance.

Hypothesis testing

The study currently uses a combination of exploratory and hypothesis-driven approaches. For example, directional predictions are tested regarding the effect of phylogenetic position on attitudes towards animals (p. 7), while predictions are not made for the effects of all demographic variables. I appreciate that the current study is challenging because some predictions are difficult to specify a priori, but the authors need to explain and justify their approach much more explicitly, stating which analyses are exploratory and which driven by prior predictions in advance. As a consequence of this mixed, partly exploratory approach, the authors end up having to report both main effects from ANOVAs as well as long lists of Bonferroni-corrected individual comparisons in the results section, the importance of which

is often unclear. As previously suggested (and demonstrated by the authors in their response, but not implemented in the paper), a clustering approach would be a far more efficient means of exploratory analysis of variation in attitudes to animals by species and purpose.

The mix of exploratory and hypothesis-driven approaches has been justified p7-8. We presented the results of the cluster analyses in the R1 response document but not the paper because they were found to add little. The cluster analyses confirmed a cluster for the APQ ratings (dog, dolphin, chimp together) but there was little differentiation between the other 9 animal species. The BAM output was more differentiated showing three distinct clusters. However, we are still of the opinion that such an analysis focused on similarity (as the obverse of difference between species) does not add to the manuscript. Moreover, we do not wish to suggest a classification of animals (of which we sample just a few selected species) based on BAM or attitudes to their use (Clatworthy et al., 2005. The use and reporting of cluster analysis in health psychology: A review. British Journal of Health Psychology, 10, 329-358). The dendrogram output (using average linkage between groups) is shown below for (A) the APQ and (B) the BAM scores.

(A)

(B)

In the Discussion (p23) we now acknowledge the potential value of cluster analysis for future studies which examine a wider range of species.

Similarly, the authors need to much more explicitly clarify predictions in relation to demographic variables. Although directional predictions are made only in relation to gender and diet, the authors cite several studies in the Introduction showing effects of other demographic variables (e.g. age, religion, culture) on attitudes towards animal use (p. 4). It is not currently clear why directional predictions could not be made for these variables as well.

We anticipated that the average age of the participants would be low and in a relatively restricted range. As the results turned out, the age range examined in the present study was 18-80 years but the mean age was relatively low at 38 years (SD=15.98 years) so we can't say too much about the attitudes of older participants. Effects of religion and culture are complex and the sampling strategy was also unlikely to result in sufficient diversity to examine these effects systematically (see Table 1). As explained p7, the purpose of questions pertaining to age, ethnicity, religion and level of education was in order to consider whether the sample was representative. We don't wish to present all possible exploratory analyses and some would be inevitably underpowered and/or insufficiently detailed. For example, the sample was 62.8% white and ethnicities other than Asian were reported at 5% or less. Similarly, the sample was 41% atheist and religions other than Christianity and Hinduism were reported at 3.5% or less. This point is picked up in the Discussion p19.

Collapsing scores

In order to explore the interaction between species and purpose as drivers of attitudes towards animal use, the authors collapse scores across species to compare purpose categories, and collapse scores across purpose to compare species categories (p. 12). This is unnecessary and results in a loss of statistical power; multi-level models could be used instead to avoid this and maintain greater power.

We have not been in the position to invest too much time in multi-level modelling options. However, looking into these it seems that the power for level 2 effects depends on the number of groups and that multilevel models add little over classical models in the case of < 5 groups (Gelman and Hill, 2007, Data Analysis Using Regression and Multilevel/Hierarchical Models, CUP, Chapter 11). Moreover, as explained above it's not clear what our level 2 groupings should be. The list of species was neither definitive nor exhaustive and the results of the cluster analyses were inconclusive.

Since we have followed the suggestion to provide the data also in SPSS format, and will also make the data available in that format in the University of Nottingham Research Data Repository post-publication, the data will be freely available for further analyses.

In the Discussion (p23) we now acknowledge the potential value of multilevel modelling approaches for future studies which examine a wider range of species in a nested structure.

Minor issues

- Why is it necessary to use t-tests to calculate significance values for the effects of predictors from multiple regression analyses? (p. 12/13)

These tests (the results of which are reported p17) are not strictly necessary but are useful to confirm the consistency of effects across species. In line with the Subject Editor's comments we've retained this short section of the results and the associated Table 4. The justification for these comparisons has been improved p13.

- It would be helpful to share the SPSS output file as well as the raw data with Reviewers so we can confirm details of the analysis, data processing, etc.

We have followed the suggestion to provide the data also in SPSS format and will also make the data available in that format in the University of Nottingham Research Data Repository post-publication. Thus the data will be freely available (and in a convenient format).

- The authors do not explain what sphericity is or why it is important to check (p. 13).

This is fairly standard statistics procedure. Nonetheless a citation (Girden, 1992) has been provided to explain this test for the assumptions of ANOVA and the use of the Greenhouse-Geisser correction.

- Predictions regarding the effects of phylogenetic position should still hold despite other factors, if the effect is strong enough. If the authors wish to make specific predictions regarding moderators of this effect, these could be tested explicitly in the multiple regression models (p. 7)

As explained in the Intro p7, we don't share the view that there's a compelling rationale to base predictions on phylogenetic placement. Moreover, whilst the species were selected to include examples likely to be seen as familiar and to fall into the categories of pet, pest and profit (as explained p6), this was not intended to be an exhaustive list, nor would we claim that the species list was definitive. The rationale for the 'blueprint' (p8) or template (p23) approach has been further improved in response to Reviewer 1.

- The rule of thumb used for sample sizes is not sufficiently justified – the number 104 seems completely arbitrary (p. 15).

The '104+m' rule of thumb, for the suggested minimum number of participants in relation to the number of predictors (m), for regressions is explained in the reference entitled 'How many subjects does it take to do a regression analysis' which is cited new p16 (Green, 1991). There are also more complex rules of thumb which are evaluated within the cited article but '104+m' is already an improvement on $N > 100$ and our samples sizes are in the range $N=272-317$ and so well above what the available rules of thumb suggest.

- Phylogenetic 'scale' should be avoided in favour of a neutral term on p. 19

Apologies for the oversight, this has been corrected (and the further searches for phylogenetic status and scale confirm that none remain).